# Repurposing a chemosensory macromolecular machine

Davi R. Ortega [1], Wen Yang[2], Poorna Subramanian[1], Petra Mann[3], Andreas Kjær [1,9], Songye Chen[1], Kylie J. Watts [4], Sahand Pirbadian[5], David A. Collins[6], Romain Kooger[7], Marina G. Kalyuzhnaya[6], Simon Ringgaard[3], Ariane Briegel [2✉] & Grant J. Jensen [1,8✉]

How complex, multi-component macromolecular machines evolved remains poorly understood. Here we reveal the evolutionary origins of the chemosensory machinery that controls flagellar motility in *Escherichia coli*. We first identify ancestral forms still present in *Vibrio cholerae*, *Pseudomonas aeruginosa*, *Shewanella oneidensis* and *Methylomicrobium alcaliphilum*, characterizing their structures by electron cryotomography and finding evidence that they function in a stress response pathway. Using bioinformatics, we trace the evolution of the system through γ-Proteobacteria, pinpointing key evolutionary events that led to the machine now seen in *E. coli*. Our results suggest that two ancient chemosensory systems with different inputs and outputs (F6 and F7) existed contemporaneously, with one (F7) ultimately taking over the inputs and outputs of the other (F6), which was subsequently lost.

[1] Division of Biology and Biological Engineering, California Institute of Technology, 1200 E. California Blvd, Pasadena, CA C1125, USA. [2] Institute of Biology, Leiden University, 2333 BE Leiden, The Netherlands. [3] Department of Ecophysiology, Max Planck Institute for Terrestrial Microbiology, D-35043 Marburg, Germany. [4] Division of Microbiology and Molecular Genetics, School of Medicine, Loma Linda University, Loma Linda, CA 92350, USA. [5] Department of Physics and Astronomy, University of Southern California, Los Angeles, CA 90089, USA. [6] Department of Biology, Viral Information Institute, San Diego State University, San Diego, CA 92182, USA. [7] Institute of Molecular Biology and Biophysics, Eidgenössische Technische Hochschule Zürich, CH-8093 Zürich, Switzerland. [8] Howard Hughes Medical Institute, California Institute of Technology, Pasadena, CA 91125, USA. [9]Present address: Rex Richards Building, South Parks Road, Oxford OX1 3QU, UK. ✉email: a.briegel@biology.leidenuniv.nl; jensen@caltech.edu

Cells are full of complex, multi-component macromolecular machines with amazingly sophisticated activities. In most cases, how these machines evolved remains mysterious. Presumably, they arose through a long series of small steps in which new components and functions accreted onto, or replaced, original ones. Throughout this process, each new function provided a fitness advantage and was thus retained. The chemosensory pathway in bacteria and archaea is one such multi-component system. It integrates environmental signals to control cellular functions ranging from flagellum- and pilus-mediated motility to biofilm formation. Also, chemosensory proteins are key virulence factors for many pathogens. The best-understood function of chemosensory systems is their control of the rotational bias of the flagellar motor, guiding bacteria toward attractants and away from repellents[1,2].

The molecular basis of this activity has been the object of intense study in *Escherichia coli*, where transmembrane methyl-accepting chemotaxis proteins, or MCPs, form large arrays at the cell pole[3]. These chemoreceptors bind attractants or repellents in the periplasm and relay signals to a histidine kinase (CheA) in the cytoplasm[4]. When activated, CheA first autophosphorylates and then transfers the phosphoryl group to the response regulators CheY and CheB, a methylesterase. Phosphorylated CheY binds to the flagellar motor, changing the direction of flagellar rotation. This allows the cells to switch from swimming forward smoothly (so-called runs) to tumbling randomly. Changes in the duration and frequency of run and tumble phases drive a biased random walk that moves the cells towards favorable environments[5]. The signal is terminated by a phosphatase, CheZ, that dephosphorylates free CheY[6]. Phosphorylated CheB tunes the sensitivity of the system by changing the methylation state of the chemoreceptors, opposing the constitutive activity of the methyltransferase CheR[7,8].

While the chemosensory system in *E. coli* is well understood, the structure and function of many others is not. Chemosensory systems have been classified on the basis of evolutionary history into 17 so-called flagellar classes (F1–17), one type IV pili class (TFP) and one class of alternative cellular functions (ACF)[9]. Because this classification system is based on phylogenomic analysis, the evolutionary relationship between the classes is generally known. Later, by analyzing chemosensory systems in archaeal genomes, we showed evidence that class F1 is the most ancient of the chemosensory classes[10]. Understanding this classification system and its evolution allows for a temporal directionality of the evolution and diversification of this system. However, the class names are not reliable predictors of biological role. In *E. coli*, the system that controls the flagellar motor is a member of the F7 class, but in many other bacteria this is not the case. Conversely, in *Rhodospirillum centenum* a member of the F9 class controls biosynthesis of flagella[11]. Historically, all these pathways have been called chemotaxis pathways in reference to their homology to the biological pathway that gives rise to the chemotaxis phenotype in a diverse set of organisms including *E. coli*. Here, we will refer to them instead as chemosensory pathways, to reflect the diversity of outputs that these pathways modulate in response to chemical cues in the environment.

In previous work, we and others have used electron cryotomography (cryo-ET) to reveal the in situ macromolecular organization of several chemosensory systems[8–10,12–14]. This method allows the study of bacterial cells in a near-native state in three dimensions at macromolecular resolution. Cryo-ET revealed that all the chemosensory systems controlling flagellar motors that have been imaged so far, including the F6 systems of various γ-proteobacteria and the F7 system of *E. coli*, look very similar[12]. Here, imaging some of these same species under stress, we observed a new kind of chemosensory array. Surprisingly, we

identify it as another form of F7, but with a remarkably different structural architecture compared to that of the canonical *E. coli* F7 system. Tracing its evolutionary history, we find that this unusual F7 system actually represents the ancestral form, which in a series of defined steps acquired both the input and output domains of the ancient F6 system to take over control of the flagellar motor, leading to the system seen in modern *E. coli*. The result is a fascinating example of the evolutionary repurposing of complex cellular machinery.

## Results

**Unusual chemosensory array in *V. cholerae* and *P. aeruginosa*.** Previously, we used cryo-ET to reveal the structure of two types of chemosensory arrays in *V. cholerae*. When grown in rich medium, the cells contain a polarly-localized membrane-bound array, with a distance of 25 nm between the inner membrane (IM) and the baseplate, composed of kinase and scaffold proteins[12,15]. We showed that this array is formed by proteins of the F6 chemosensory system[16] (known to control flagellar rotation). In late stationary phase, we found that cells contain another, purely cytoplasmic array consisting of two CheA/CheW baseplates 35 nm apart sandwiching a double layer of chemoreceptors[16]. This array is formed by proteins of the F9 chemosensory system, but its function is unknown. Here, imaging cells grown into late stationary phase, we observed a third array type. This third type, present in 35% of cells in late stationary phase (Supplementary Table 1), was membrane-associated and located at the cell pole near the F6 arrays, but was taller than F6 arrays, with a distance between the IM and CheA/CheW baseplate of $38.4 \pm 1.9$ nm (Fig. 1A).

Imaging another γ-proteobacterial species, *Pseudomonas aeruginosa*, grown in nitrogen-limited media, we again observed both short and tall membrane-bound arrays. The short arrays were located at the cell poles, typically in close proximity to the single flagellar motor. The distance between the IM and the CheA/CheW baseplate was $24.3 \pm 1.8$ nm. MCPs are classified by length according to the number of heptads (sets of seven consecutive amino-acids) that the receptors contain in their signaling domain[17]. The length of the shorter array (24 nm) corresponds to receptors belonging to the class with 40 heptads (40H) that are often associated with F6 systems[9,18], so we assign this array as the F6 system. The additional taller membrane-associated array, present in ~30% of the cells (Supplementary Table 1), was often (but not always) found at the same cell pole as the putative F6 array and had a distance of $40.3 \pm 1.8$ nm between the IM and the CheA/CheW baseplate (Fig. 1B).

**The unusual array is dependent on proteins of the F7 system.** To determine which proteins form the unusual arrays we observed by cryo-ET, we examined the genomes of *V. cholerae* and *P. aeruginosa*. The genome of *V. cholerae* encodes three chemosensory systems: F6, F7, and F9 (Supplementary Table 2)[9]. Having already identified the F6 and F9 systems[12,16], we hypothesized that the unusual arrays were formed by the F7 system. The genome of *P. aeruginosa* encodes four chemosensory systems: F6, F7, ACF, and TFP (Supplementary Table 2)[9]. Since the height of the short array matches that of receptors of the F6 system, and since arrays corresponding to ACF and TFP systems have never been observed in any organism (it is possible that they do not form complexes large enough to be visible in electron cryotomograms), we again hypothesized that the tall array in this organism is formed by the F7 system.

To test this idea, we imaged deletion mutants of F7 genes in *V. cholerae* and *P. aeruginosa* by cryo-ET. Deletion of *cheA* and/or *cheW* of the F7 gene cluster resulted in the absence of the unusual

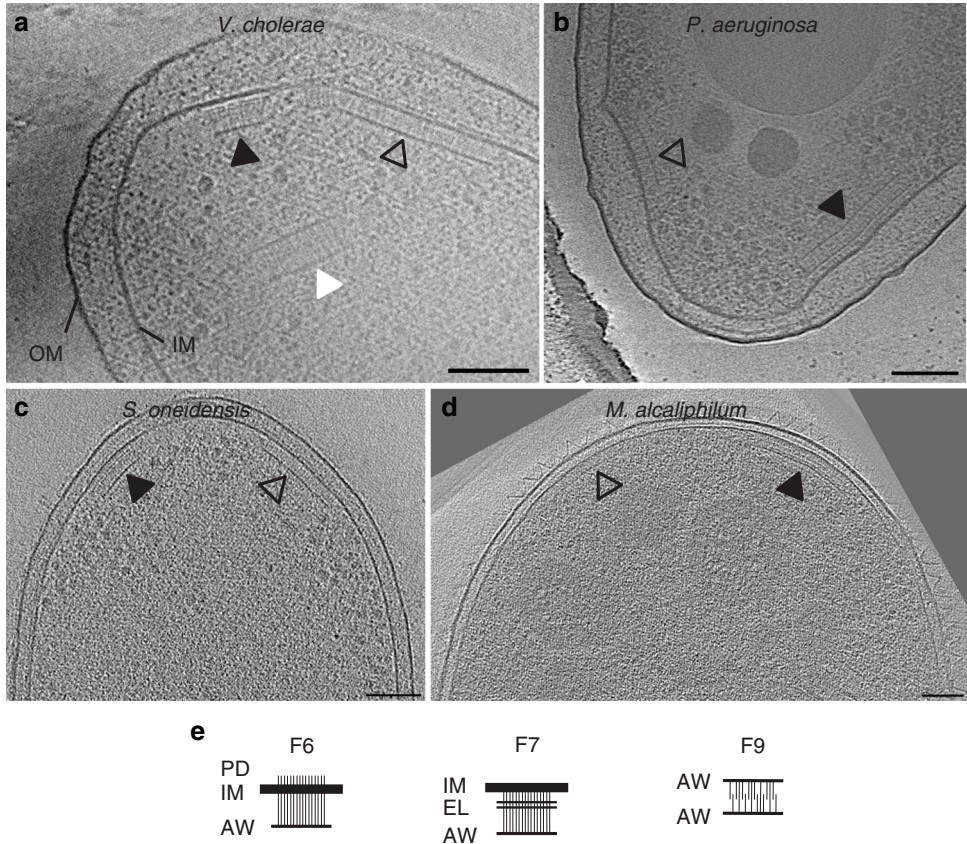

**Fig. 1 Electron cryotomography of chemosensory arrays.** Different putative chemosensory classes, F6 (empty arrows), F7 (black arrows), and F9 (white arrow), have different architectures when observed in side view in various γ-proteobacterial species: **a** *V. cholerae*, **b** *P. aeruginosa*, **c** *S. oneidensis*, and **d** *M. alcaliphilum*. Scale bars are 50 nm. **e** Diagrams of macromolecular features characteristic of the F6, F7, and F9 arrays. The F6 arrays span the inner membrane (IM) and have visible periplasmic domains (PD), and a CheA/CheW layer (AW). The F7 arrays are also membrane-bound but lack a PD. Instead, they have extra layers (EL) between the IM and the AW layer. The F9 cytoplasmic array is not bound to the membrane. Instead, the receptors are sandwiched between two AW layers.

tall array (Supplementary Table 1), consistent with our hypothesis that these arrays correspond to the F7 chemosensory system. In both organisms the F7 gene cluster contains two MCPs: one presumably cytosolic class 36H receptor (Aer2/McpB/PA0176 in *P. aeruginosa* and Aer2/VCA1092 in *V. cholerae*), and one receptor of uncategorized class with a predicted transmembrane region (Cttp/McpA/PA0180 in *P. aeruginosa* and VCA1088 in *V. cholerae*). In both species, deletion of the McpA-like receptor had no effect on the presence of the unusual array, but deletion of the Aer2 receptor abolished the array (Supplementary Table 1). We therefore conclude that Aer2 receptors, and not McpA-like receptors, are required for the formation of the F7 arrays.

**F7 chemosensory systems are widespread in γ-proteobacteria.** The histidine kinase CheA gene is used as a proxy to find major chemosensory clusters in genomes[9]. We selected a non-redundant set of 310 γ-proteobacteria genomes containing at least one CheA, and found that more than half (176) contained at least one F7 CheA. None of the species we analyzed had more than one F7 system. In the course of other projects, our group has used cryo-ET to image many bacterial species; we found that two of the species in our imaging database[19]—*Shewanella oneidensis* MR-1[20] and *Methylomicrobium alcaliphilum* 20Z[21]—were γ-proteobacteria with F7 systems.

The genome of *S. oneidensis* contains two chemosensory systems, one from the F6 class (SO_3200-SO_3209) and another

from the F7 class (SO_2117-SO_2126, Supplementary Table 2)[9]. In cryotomograms of *S. oneidensis* cells grown anaerobically in batch culture, we observed a single membrane-bound array, usually located at the cell pole in close proximity to the single flagellar motor. The distance between the IM and the CheA/CheW baseplate was $24.5 \pm 2.7$ nm, as expected for 40H chemoreceptors of the F6 system[18]. When cells were grown anaerobically in continuous flow bioreactors, however, we observed the unusual taller array type in ~10% of cells, often (but not always) at the same cell pole as the F6 array (Supplementary Table 1). This array had a distance of $35.5 \pm 2.7$ nm between the IM and the baseplate (Fig. 1C).

Chemotaxis in *M. alcaliphilum* has yet to be explored, but genome analysis predicts three chemosensory gene clusters, one each from the F6 (MEALZ_3148 - MEALZ_3158), F7 (MEALZ_2869-MEALZ_2879) and F8 classes (MEALZ_2939–MEALZ_2942, Supplementary Table 2). Cryo-ET of *M. alcaliphilum* revealed two array types: putative F6 arrays with $25.6 \pm 2.8$ nm between the IM and the CheA/CheW layer, as expected for 40H chemoreceptors[18], and, in 25% of cells, taller arrays with a distance of $35.1 \pm 2.8$ nm between the IM and baseplate (Fig. 1D). The F8 chemosensory system uses a class 34H chemoreceptor with two transmembrane regions. Given the domain architecture in the cytoplasmic portion of the sequence, we expect arrays formed by these receptors to exhibit a distance of ~22 nm between the IM and the CheA/CheW baseplate[12]. We did not observe any such array in our cryotomograms. We therefore

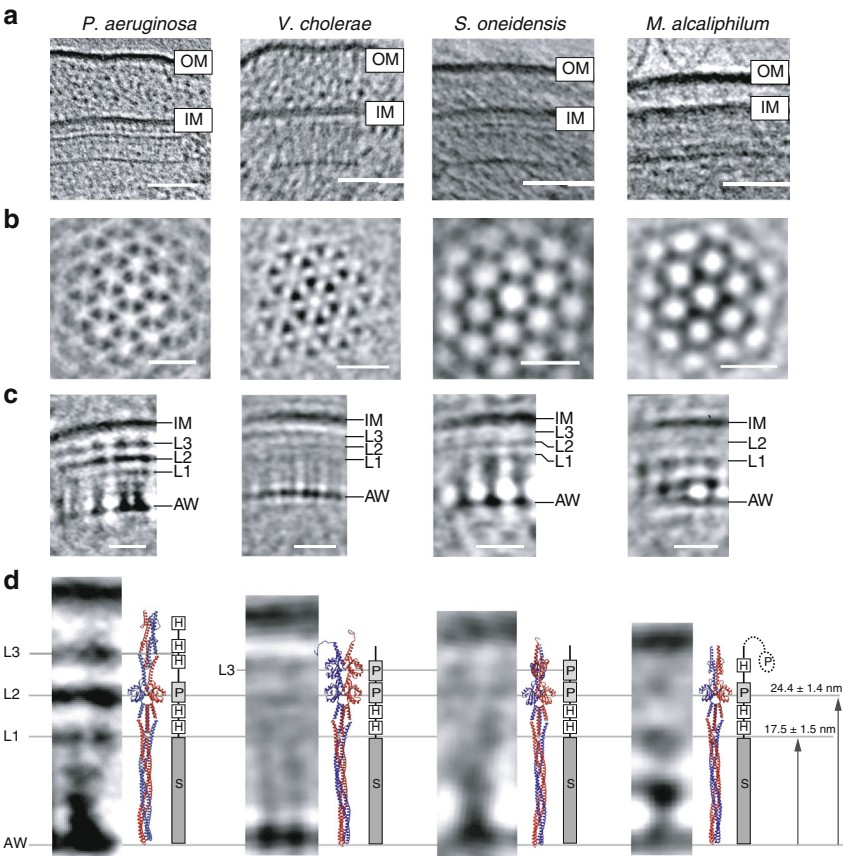

**Fig. 2 Structural analysis of F7 chemotaxis arrays. a** Side-view of F7 chemotaxis arrays (scale bar: 50 nm) relative to the inner membrane (IM) and outer membrane (OM). **b** Top-view of a sub-tomogram average at the CheA/CheW layer reveals that F7 arrays have the typical hexagonal packing of receptors with ~12 nm spacing (scale bar: 20 nm). **c** Side view of the sub-tomogram averages. Several density layers (L1–L3) are present between the inner membrane and the CheA/CheW layer (AW) (scale bar: 20 nm). **d** A comparison of the side-views from the sub-tomogram averages with homology models of chemoreceptors (to scale). The presence of PAS domains in the receptor structures consistently correlate with L2 layer in all organisms and the L3 layer in *V. cholerae* and *S. oneidensis*. The box diagram shows the PFAM protein domains of each receptor: PAS (P), HAMP (H), and MCPsignal (S). The distance measurements were taken using 1D average profile and the uncertainty reported as described in the Methods section.

assume that the taller arrays are F7. We conclude that the tall F7 arrays are widespread across γ-proteobacteria based on two results: (1) the bioinformatics analyses indicate the presence of homologs of the proteins forming tall F7 arrays in nearly all γ-proteobacteria (except the enterics) and (2) the presence of tall arrays in tomograms of all four species with predicted F7 systems in our database.

**F7 array architectures match domains of Aer2-like receptors.** In typical F6-like membrane-bound arrays, including all those imaged by cryo-ET in this and previous studies, a layer of periplasmic domains is visible just outside the IM[12]. In contrast, the unusual F7 arrays lacked discernable periplasmic densities. Instead, they exhibited multiple cytoplasmic layers between, and parallel to, the IM and the CheA/CheW baseplate. To better visualize these additional layers, we computed 3D sub-tomogram averages as well as 1D profiles of the F7 array in each species (Fig. 2, Supplementary Table 3). All strains contain additional density layers between the CheA/CheW layer and the IM. We labeled them in order from the CheA/CheW layer towards the IM as L1, L2 in *M. alcaliphilum* and L1–L3 in *P. aeruginosa*, *V. cholerae*, and *S. oneidensis*, respectively (Fig. 2C).

We wondered whether these density layers we observed by cryo-ET in the unusual F7 array corresponded to structural features of its Aer2-like receptor. The *P. aeruginosa* Aer2 receptor

is well characterized: it consists of three N-terminal HAMP domains, followed by a PAS domain, two additional HAMP domains and a cytoplasmic signaling domain[22,23]. The *V. cholerae* Aer2 receptor is also well characterized: it consists of two N-terminal PAS domains, two HAMP domains and a cytoplasmic signaling domain[24]. We used CD-VIST[25] to predict the domain architecture of the related receptor in the two remaining species. In *S. oneidensis*, the Aer2-like receptor consists of two N-terminal PAS domains, followed by two HAMP domains and a cytoplasmic signaling domain, like Aer2 from *V. cholerae*. As in *P. aeruginosa*, the receptor lacked any discernable transmembrane region and was predicted to be cytosolic. The *M. alcaliphilum* Aer2-like receptor was predicted to contain an N-terminal PAS domain followed by a single HAMP domain, another PAS domain, two HAMP domains and finally a cytoplasmic signaling domain. Unlike in the other species, though, this receptor was predicted to contain two short potential transmembrane regions (10 and 14 residues) between the N-terminal PAS domain and the rest of the protein. PAS domains are rarely periplasmic[26], however, suggesting that even if these regions are embedded in the membrane, they do not span it.

Using this information, we constructed a homology model of the Aer2-like receptor in each species based on available atomic models of the protein domains and assuming that the individual domains stack linearly within the four-helix bundle of the receptor dimer. We then manually aligned the homology model

for each organism with the corresponding electron density profile (Fig. 2D). In all four cases, the receptor fit well between the CheA/CheW baseplate and IM. We also observed a correlation between densities observed by cryo-ET and domain features of the receptors. In all four species, the first layer, L1, corresponded to the boundary between the cytoplasmic signaling domain and its proximal HAMP domain $17.5 \pm 1.5$ nm above the CheA/CheW baseplate. The L2 layer $24.5 \pm 1.4$ nm above the CheA/CheW baseplate corresponded to the PAS domain present in all four species. The second PAS domain present in *V. cholerae* and *S. oneidensis* approximately correlated with the L3 layer $30.1 \pm 2.3$ nm above the CheA/CheW baseplate. *P. aeruginosa* does not have a second PAS domain; its L3 layer instead appeared to match a HAMP domain ($30.7 \pm 1.8$ nm). The distances from the CheA/CheW baseplate to each layer for each species are listed in Supplementary Table 4. These correlations further support the conclusion that the unusual tall arrays are formed by the Aer2-like MCPs of the F7 system and that some of these layers consistently correlate with the PAS domains of the receptors in each species (Fig. 2D).

**Evolution of the F7 system exhibits distinct stages**. These findings present a puzzle: the F7 arrays in *P. aeruginosa*, *V. cholerae*, *S. oneidensis*, and *M. alcaliphilum* all look similar to one another, but very different than the F7 systems we had observed previously in *E. coli* and *S. enterica*. Instead, the *E. coli* and *S. enterica* F7 systems resemble the F6 systems we saw in *P. aeruginosa*, *V. cholerae*, *S. oneidensis*, and *M. alcaliphilum*. This prompted us to explore the evolutionary relationship between these systems.

To do that, we first constructed a phylogenetic tree of proteobacterial F7 systems using concatenated sequences of CheA, CheB, and CheR (Fig. 3, Supplementary Fig. 1). To give a temporal direction of evolution to this analysis, we included sequences from F8 systems to help root the tree because it is unlikely that either one of these classes is the most ancient class of chemosensory systems[9,10]. To track the evolution of the system, we grouped monophyletic branches into clades.

The first clade comprised all of the F7 from ε-proteobacterial, most of F7 from δ-proteobacterial and all of the F8 systems, the second all the α-proteobacterial systems, the third and fourth nonenteric γ-proteobacterial systems, the fifth and sixth β-proteobacterial systems, and the seventh the enteric γ-proteobacteria. The F7 chemosensory systems of the organisms from the same proteobacteria class tend to cluster together. To track the distribution of the F7 system through these clades, we built a phylogenetic profile using a new random set of 161 γ-proteobacteria, the four species imaged in this work, the model organism *E. coli*, and ten β-proteobacteria to serve as an outgroup (Supplementary Fig. 2). The complete list of genomes is shown in Supplementary Table 10, and the phylogeny was built using 31 orthologs according to the protocol by Ciccarelli et al.[27].

Next, we analyzed the arrangement of genes in the F7 gene cluster in γ- and β-proteobacterial species. The gene arrangement of F7 systems significantly differ from that of the F6 systems[9]. We found a specific and characteristic organization of the F7 gene cluster in each of the last five clades of the phylogenetic tree. For clarity, we will refer to these as the first through fifth evolutionary "stages" of the F7 system. The most ancient of these groups, stage 1, was marked by the presence of an anti-sigma factor antagonist followed by *cheY*, *cheA*, two *cheW*-like genes, an *aer2*-like chemoreceptor, *cheR*, *cheD*, *cheB*, an *mcpA*-like chemoreceptor, another anti-sigma factor antagonist and a serine phosphatase. All organisms with a stage 1 F7 system also contained an F6 system elsewhere in their genome (Fig. 3, Supplementary

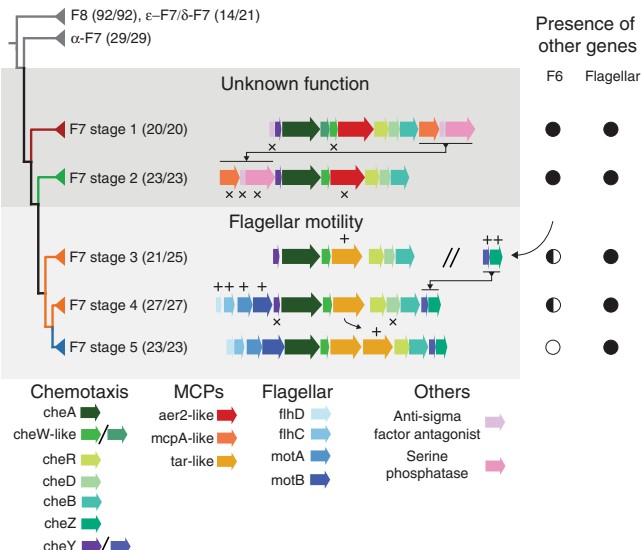

**Fig. 3 Major events in the evolutionary history of the F7 system in γ-Proteobacteria.** Each major branch is identified by the type of its F7 system, and shading indicates the function of the system. Numbers in parentheses indicate the number of genomes from the designated class compared to the number of genomes present in the branch. Differences in these numbers indicate lateral gene transfers and are highlighted in Supplementary Fig. 1. The double division symbol is used to show that the protein clusters on each side of the symbol are located in different parts of the genome, the symbols plus and crosses indicate gene additions and losses, respectively. The presence and absence of other relevant systems in the genomes containing each stage are marked as complete (full circle), partial (half circle), or absent (empty circle).

Fig. 2). In the transition to stage 2, the most 5' anti-sigma factor antagonist of the gene cluster and one of the *cheW*-like genes were lost and the last three downstream genes moved to the front of the chemosensory cluster. Again, all organisms with a stage 2 F7 system had a complete F6 system. Next, in stage 3 these same three (now upstream) genes, including the *mcpA*-like receptor, were lost and an *aer2*-like receptor gene was replaced by a chemoreceptor gene with two transmembrane regions, a periplasmic sensory domain like those found in well-studied model receptors in *E. coli* (Tar, Tap, Trg, and Tsr) and lacking other cytoplasmic protein domains except HAMP and MCP signaling (Supplementary Fig. 3). In this transition, changes also occurred in the F6 gene cluster: only seven of 21 stage 3 genomes still had an F6 CheA, and none had other core proteins like CheB and CheR, suggesting that the F6 CheA may no longer be functional. All species, however, maintained the F6 *cheY*/*cheZ* pair somewhere in the genome. In stage 4, four flagellar genes (*flhD*, *flhC*, *motA*, and *motB*) moved to the front of the F7 cluster and the *cheY*/*cheZ* pair previously associated with the F6 system moved to the back. As a result, the F7 cluster now had two *cheY* genes: one from the original F7 system and another from the F6. Further losses occurred in F6 genes: only 3 of 27 stage 4 genomes retained the F6 CheA (also with no F6 CheB or CheR). Finally, in stage 5 (where enteric γ-proteobacteria like *E. coli* emerged), *cheD* and the more upstream F7 *cheY* were lost, and the *tar*-like chemoreceptor gene was duplicated. None of the stage 5 genomes retained any genes from the F6 system. For reference, the species imaged in this study have F7 systems from stage 1 (*V. cholerae* and *M. alcaliphilum*) and stage 2 (*P. aeruginosa* and *S. oneidensis*).

The flagellar motor is controlled by the F6 system in many species with stage 1 or 2 F7 systems[28–30], and it is controlled by

the F7 system in stage 5 species (the enterics). Our results therefore suggest that control of the motor switched from the ancestral F6 system to the tall F7 system in the transition between stages 2 and 3. Less is known about stage 3 and 4 species (β-proteobacteria), but at least some use flagella and have been reported to be chemotactic[28]. Based on our results, we predict that the F7 system controls the flagellar motor in these organisms. The function of the F7 system in stages 1 and 2 remains unclear.

**Evolution of the system's inputs**. How did the F7 system take control of the flagellar motor? To address this question, we examined both the inputs and outputs of the system. First, we examined the inputs by analyzing the chemoreceptors. In stages 1 and 2, the F7 cluster contained at least one *aer2*-like receptor gene from the heptad class 36H as well as an *mcpA*-like gene. The F6 gene cluster does not contain a chemoreceptor. However, 40H receptors with a topology similar to *tar*-like genes that are known to work with F6 systems in several nonenteric γ-proteobacteria are present in these genomes. Because nearly all the genomes with stage 1 and 2 F7 systems possess *mcpA*-like and *aer2*-like chemoreceptor genes in the gene cluster, we hypothesize that both receptors are needed for the yet-unknown function of F7 in these organisms. This is puzzling, however, since as described above, we found that McpA-like receptors were nonessential for formation of the F7 array. Suggesting an alternative scenario, a previous study found that F7 McpA-like receptors physically associate with the F6 array[29].

Of the 26 chemoreceptors present in the genome of *P. aeruginosa*, only two receptors present in the F7 system gene cluster, McpA and Aer2, have a characteristic C-terminus extension[31]. In the case of Aer2, this extension ends with a particular pentapeptide motif that is known to tether CheR2 (which is also part of the F7 gene cluster), and enhances its enzymatic activity[31]. Our bioinformatics analysis shows that out of the 130 Aer2-like receptors in our dataset, 96 contained a matching pentapeptide tether motif: -x-[HFYW]-x(2)-[HFYW][17] (Supplementary Table 11). Aer2-like sequences lacking such a peptide tether were exclusively found in genomes that contained at least one other Aer2 homolog containing a peptide tether. Therefore, we conclude that this motif represents a fundamental feature of the Aer2-like family.

In contrast, the C-terminus of McpA in *P. aeruginosa* does not possess the pentapeptide motif. Instead, a valine occupies the second position of the motif. This replacement prevents an interaction with any of the 4 CheR homologs[31]. Analysis of the other 39 McpA-like sequences in our dataset revealed that none of them possessed the -x-[HFYW]-x(2)-[HFYW] motif. However, the C-terminus of McpA-like sequences is highly conserved (Supplementary Fig. 4 and Supplementary Table 12). The striking conservation of this region among all the McpA sequences suggests that this motif serves an important yet unclear biological role.

To explore the evolutionary history of these two receptors, we identified 130 Aer2-like and 39 McpA-like chemoreceptors from the pool of 166 γ-proteobacterial genomes we used to build the phylogenetic profile in Supplementary Fig. 2 (some Xanthomonadales contained multiple copies of Aer2-like receptors in their F7 gene clusters, some Shewanellaceae lacked McpA-like receptors and none of the Enteric genomes had either). All 130 identified Aer2-like receptors belonged to the 36H heptad class, consistent with their belonging to F7 systems[9,18]. The McpA-like receptors could not be assigned to a heptad class. Inferring the relationships among these receptors with a phylogenetic tree, we found that the evolutionary histories of both Aer2- and McpA-like chemoreceptors are largely congruent with that of the CheABR

phylogeny (Supplementary Fig. 5). More specifically, they recapitulate the split between stage 1 and 2 F7 systems. Proteins often co-evolve when they participate in the same, or codependent, biological functions, so this again suggests that both McpA-like and Aer2-like receptors mediate the ancestral F7 function. This was expected for Aer2-like receptors (which are part of the F7 array), but surprising for McpA-like receptors (which are part of the F6 array). It is unclear what function McpA-like receptors might perform for the ancestral F7 system while physically integrating into the F6 array.

The change in the biological function of the F7 system apparently coincided with the change of the *aer2*-like gene to a *tar*-like gene. To investigate this switch, we looked further into the domain architecture of the Aer2- and Tar-like MCPs. While their signaling domains are similar (all belong to the 36H heptad class), the rest of their topology differs. As described above, Aer2-like receptors have multiple PAS-HAMP repeats and no periplasmic domain (Fig. 2D). In contrast, 36H F7 Tar-like receptors have a periplasmic N-terminal sensor sandwiched by two transmembrane regions and a single HAMP domain[30]. Interestingly, this topology is the same as that of the majority of 40H F6 receptors that control flagellar motility in γ-proteobacteria with stage 1 and 2 F7 systems[18]. Thus, the F7 system's takeover of the flagellar motor involved acquisition of a sensory input (an N-terminal periplasmic sensor domain) similar to that of the F6 system used to control the motor.

**Evolution of the system's outputs**. Finally, we examined the outputs of the system: *cheY* and *cheZ*. We first assigned *cheY* and *cheZ* genes to F7 and F6 systems based on their location in gene clusters (Fig. 3). We observed that organisms with stage 1 or 2 F7 systems possessed *cheY* genes in their F7 and F6 clusters, but only the F6 systems also included *cheZ*. The *cheY*/*cheZ* pair from the F6 systems was retained in stages 3 and 4, even as the remainder of the F6 cluster was lost. The ancient *cheY* gene from the F7 system was finally lost in stage 5. To test whether the sequences support this history, we performed a phylogenetic analysis of all *cheY* genes present in the 246 proteobacterial genomes used for the CheABR analysis (Fig. 4). This shows that the "extra" *cheY* genes outside the F7 gene cluster in stage 3 genomes were more closely related to F6 *cheY* genes than to the *cheY* genes present in F7 stage 1 and 2. This F6-related *cheY* is the same one that appears at the downstream end of the F7 cluster in stage 4, and also the one that remains in stage 5 (as the old F7-related *cheY* gene near the front of the cluster is lost). It was previously shown that the *cheZ* genes that moved into the F7 cluster were descendants of the F6 *cheZ*s[9]. This supports the notion that as the F7 system took over control of the flagellar motor, it lost its original output and acquired the motor-controlling outputs of the F6 system.

Based on our hypothesis, we predict that CheY had to acquire mutations in order to switch its interaction partner to CheA-F7. To identify these mutations, we selected the CheY protein sequences from CheY-F6 in organisms with stages 1 or 2, where the complete ancestral F7 system is still present. We then compared these sequences to those of the CheY-F6-like proteins that are predicted to interact with CheA-F7 in stages 3–5. We summarized the sequence variability of each group using sequence logos (Fig. 5A). We found that ten positions were highly conserved in each group. They are the same in the groups with CheY-F6-like sequences (interacting with CheA-F7), but differ from those present in CheY-F6 (interacting with CheA-F6). We mapped these positions to the *E. coli* CheY nuclear magnetic resonance (NMR) structure (PDB: 2LP4)[32] in complex with the P1 and P2 domains of CheA (Fig. 5B). Only two of these

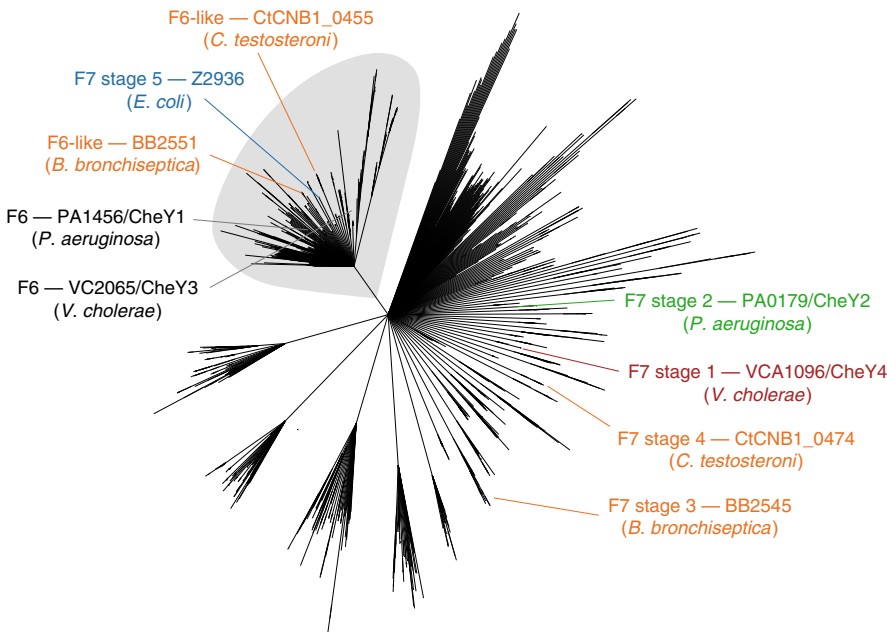

**Fig. 4 Phylogenetic reconstruction of CheY in selected proteobacterial genomes.** Sequences of CheY proteins from F6, F6-like, and F7 stage 5 systems cluster in a monophyletic group (grey shade). CheY sequences from the F7 systems of other stages (1–4) appear in other clades, indicating that F6-like and F7 stage 5 CheYs are more closely related to CheYs from F6, than F7 systems. Nodes are collapsed on 50% bootstrap support. For each stage, a representative sequence is highlighted, named with the type of its CheY, the locus number/gene name (when annotated) and the name of the organism to which it belongs. The representative genomes for each stage are: Stage 1: *Vibrio cholerae*, Stage 2: *Pseudomonas aeruginosa*, Stage 3: *Bordetella bronchiseptica*, Stage 4: *Comamonas testosteroni*, and Stage5: *Escherichia coli*. These sequence names follow the color code of stages in Fig. 3.

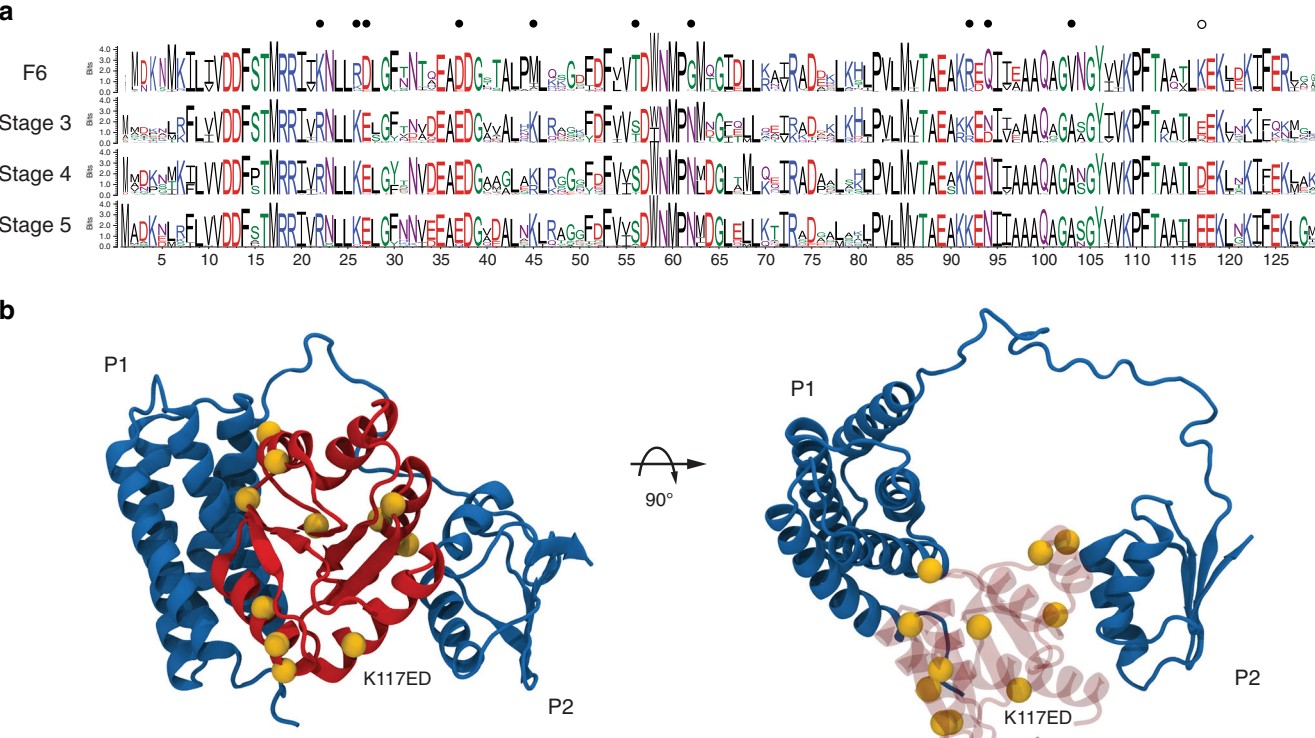

**Fig. 5 Adaptation of CheY-F6 to function with F7 systems in stages 3–5. a** Comparison of the sequence logos of the CheY proteins from the F6 system and CheY-F6-like present in genomes with 3–5 F7 systems. The amino-acids are color coded according to their chemical properties: neutral (purple), polar (green), positive charge (blue), negative charge (red), and hydrophobic (black). The dots mark conserved positions within each group that are similar between stages 3–5 but in the group of CheY-F6. **b** The alpha carbons (yellow spheres) of these positions are mapped in the structure of CheY (red) bound to the P1 and P2 domains of CheA (blue). From the ten positions that matched our criteria, only two of them were not located at an interface with CheA (M45K and T56S). Three residues are facing P2 (R92K, Q94N, and V103A) and five are facing P1 (K22R, R26K, D27E, D37E, and G62N). Note that the numbers refer to the coordinates of the *E. coli* CheY.

positions were not located at the interface with CheA domains (M45K and T56S). From the other eight residues, three face P2 (R92K, Q94N, and V103A) and five face P1 (K22R, R26K, D27E, D37E, and G62N). Furthermore, we found a charge reversal in position K117E (D in stage 4, Fig. 5B). This residue is located in the helix that is known to interact with FliM, which is the protein-binding partner of CheY in the flagellar motor[33]. Overall, these data support our hypothesis that the F7 system in stage 3 recruited the CheY/CheZ pair from the F6 system, and this only required a small set of residue changes that are mainly located in the interface with the CheA P1–P2 domains.

## Discussion

Here, using a combination of cryo-ET and bioinformatics, we have characterized and dissected the evolution of the F7 chemosensory array in γ-proteobacteria. We find that the ancient F7 system, still present in nonenteric γ-Proteobacteria, took control of the flagellar motor from the F6 system in a series of clear evolutionary steps (Fig. 6). Thus, the well-studied chemosensory model system of E. coli is a chimera of two other, more widespread systems: the F6 flagellar-control system and an ancient F7 system of still-unknown biological function. These results provide a striking example of how evolution can repurpose macromolecular complexes for new functions.

We identified four sequential evolutionary steps, each of which produced a stable, modern subtype of chemosensory array. In the step in which flagellar control moved from the F6 to the F7 system, two major evolutionary events were required: (i) the Aer2-like F7 receptor became Tar-like, swapping its input (Fig. 6B); and (ii) the F7 CheA began signaling through the remaining F6 CheY, adding an output. We speculate that this receptor transformation may have occurred via a domain swap that replaced the multiple PAS-HAMP domains of an Aer2-like receptor with the sensor domain of an F6 Tar-like receptor. These changes were accompanied by eventual loss of the remaining F6 components, as well as F7 components no longer needed for its new function (Fig. 6C). Thus, we hypothesize that intermediate stages of the F7 system, present in extant β-proteobacteria, retain both the older and younger functions. Together, these findings suggest several hypotheses on the biological role of individual components and the function of the F7 system in γ- and β-proteobacteria, see Supplementary Discussion.

## Methods

**Strains and growth conditions**. The list of Vibrio cholerae strains and construction contains the wild-type: Vibrio cholerae C6706, the strain PM6: Δvca1088, the strain PM7: Δvca1093, Δvca1094, Δvca1095 (ΔF7), and the strain PM18: Δvca1092. V. cholerae deletion strains were generated using standard allele exchange[34] with the following plasmids.

Plasmid for deletion of vca1093, vca1094 and vca1095 (pPM045) was constructed by PCR amplification of the up- and down-stream regions of vca1093 and vca1095, respectively. PCR1 was performed with primers CCCCCTCTAGA AATTGGCTAATCCCTCCTAAACTC/AATCTTGCGCAGTTGTTCCATATC and C6706 chromosomal DNA as template. PCR2 was performed with primers GATATGGAACAACTGCGCAAGATT CGCTTAAGCACCACTGCCGAA/CCC CCTCTAGACATCATCAAATTCGTCGTCATGC and C6706 chromosomal DNA as template. A third PCR was then performed using primers CCCCCTCTAGAA ATTGGCTAATCCCTCCTAAACTC/CCCCCTCTAGACATCATCAAATTC GTCGTCATGC and PCR1 and PCR2 as template. The product from PCR3 was then digested with XbaI and ligated into the equivalent site of plasmid pCVD442[34] resulting in plasmid pPM045.

Plasmid for deletion of vca1092 (pPM051) was constructed by PCR amplification of the up- and down-stream regions of vca1092. PCR1 was performed with primers CCCCCTCTAGAACGGTTGTCTTGATCTTGAGTGC/AACAAACTGGGGCAC AACCTG and C6706 chromosomal DNA as template. PCR2 was performed with primers CAGGTTGTGCCCCAGTTTGTTATGCATAAAGCACCGATAAA TCAGG/ CCCCCTCTAGAATTGCCTTGCTGATCTTTGACCC and C6706 chromosomal DNA as template. A third PCR was then performed using primers CCCCC TCTAGA ACGGTTGTCTTGATCTTGAGTGC/CCCCCTCTAGAA TTGCCTTGCTGATCTTTGACCC and PCR1 and PCR2 as template. The product

from PCR3 was then digested with XbaI and ligated into the equivalent site of plasmid pCVD442 resulting in plasmid pPM051.

Plasmid for deletion of vca1088 (pSR1228) was constructed by PCR amplification of the up- and down-stream regions of vca1088. PCR1 was performed with primers CCCCCTCTAGAAAAGCCAATGTAGGGTTTGTGCAG/ TATCGCCGTTATTTTGTGTTTTCTCG and C6706 chromosomal DNA as template. PCR2 was performed with primers CGAGAAAACACAAAATAACGG CGATAAACCGTGGGGGATTGGCTG/CCCCCTCTAGATGCGATAATGTGCC TGTACTTTG and C6706 chromosomal DNA as template. A third PCR was then performed using primers CCCCCTCTAGAAAAGCCAATGTAGGGTTTGTGC AG/CCCCCTCTAGATGCGATAATGTGCCTGTACTTTG and PCR1 and PCR2 as template. The product from PCR3 was then digested with XbaI and ligated into the equivalent site of plasmid pCVD442 resulting in plasmid pSR1228.

During plasmid and strain construction, V. cholerae and E. coli were grown at 37 °C in LB medium or on LB agar plates containing antibiotics in the following concentrations: 200 mg/mL streptomycin; 50 mg/mL kanamycin; 100 mg/mL ampicillin; 50 mg/mL carbenicillin; and 20 mg/mL chloramphenicol for E. coli, and 5 mg/mL for V. cholerae.

For cryo-ET microscopy V. cholerae cells were grown as described previously[16]: after growth in LB medium for 24 h at 30 °C with shaking, 150 µl cell suspension was diluted into 2 ml Ca-HEPES buffer and grown for an additional 16 h with shaking at 30 °C.

The P. aeruginosa mutant strains imaged in this study were acquired from the transposon mutant collection from the University of Washington, Supplementary Table 5. Wild-type P. aeruginosa PAO1 and a PAO1 aer2 deletion mutant [PAO1047][29] were imaged in this study. Cells were grown in MOPS-based nitrogen starvation medium for ~24 h at 30 °C with shaking. MOPS-based minimal medium limited nitrogen[35]: 43 mM NaCl, 50 mM MOPS (from 1 M stock of MOPS/ NaMOPS pH 7.2), 40 mM Sodium Succinate, 1 mM $MgSO_4$, 2.2 mM KCl, 0.1 mM $CaCl$, 10 µM $FeNH_4SO_4 \cdot 7H_2O$, 1 mM $NH_4Cl$, 1.25 mM $NaH_2PO_4$.

For imaging the chemosensory systems, S. oneidensis MR-1 wild-type cells were cultured using continuous flow bioreactors (chemostats) or batch cultures as previously described[20].

For imaging intracellular structures wild-type M. alcaliphilum $20Z^R$ cells were grown in a modified nitrate mineral salts medium[36] with a final pH of 9.0 consisting of: 9.9 mM $KNO_3$, 0.8 mM $MgSO_4 \cdot 7H_2O$, 13.6 µM $CaCl_2 \cdot 2H_2O$, 0.5 M g $L^{-1}$ NaCl, 2 mM $KH_2PO_4$, 2 mM $Na_2HPO_4$, 22.5 mM $NaHCO_3$, 2.5 mM $Na_2CO_3$ along with trace elements 13.4 µM $Na_2EDTA$, 7.2 µM $FeSO_4 \cdot 7H_2O$, 4.8 µM $CuSO_4 \cdot 5H_2O$, 1 µM $ZnSO_4 \cdot 7H_2O$, 0.9 µM $Na_2O_4W \cdot 2H_2O$, 0.8 µM $CoCl_2 \cdot 6H_2O$, 0.5 µM $H_3BO_3$, 0.2 µM $MnCl_2 \cdot 4H_2O$, 0.2 µM $NiCl_2 \cdot 6H_2O$, 0.2 µM $Na_2MoO_4 \cdot 2H_2O$. Cultures were grown at 30 °C in septated bottles containing 20% $CH_4$ headspace. Cell samples for cryo-ET preparations were taken at mid-exponential phase, at a cell density of $OD_{600} = 0.5$.

**Electron cryotomography**. Cells were prepared for electron cryotomography as described previously[37]. Images were collected using either a FEI G2 300 keV field emission gun microscope or a FEI TITAN Krios 300 keV field emission gun microscope with lens aberration correction (FEI Hillsboro, OR). Both microscopes were equipped with Gatan energy filters and "K2 summit" counting electron detector cameras (Gatan, Pleasanton, CA). The data collection software used to collect the tilt series was UCSFtomo[38]. The cumulative electron dose was 160 e$^-$/Å$^2$ or less for each individual tilt series. The tomograms used in this study are available in the Electron Tomography Database—Caltech[39] and their identifiers can be found in Supplementary Table 6.

**Image analysis**. CTF correction, frame alignment and SIRT reconstruction was done using the IMOD software package[40,41]. Sub-tomogram averaging was carried out with the Dynamo software package[42,43]. F7 arrays were modeled as patches of surfaces in individual tomograms, and then particles containing IM and baseplate were cropped out based on the geometry of the surface model. Information about the number of cells, particles and pixel sizes are summarized in Supplementary Table 3. Particles were aligned first based on the IM and baseplate density, and then subsequently aligned with in-plane rotations and shifts. 2D tomoslices of the averages represent the top view and the side view of the repeating unit of the F7 arrays. A side view image was used for comparison to the receptor homology models.

**Distance measurements of electron density layers**. To measure the distance between the IM and the electron density layers, we used a custom script written in Node.js. The tool is available on node package manager (npm): https://www.npmjs. com/package/sideview-profile-average. The script uses a tomogram and model points that follow the IM at a given 2D slice as input, Supplementary Fig. 6. The final average profile is representative of a 3D tomogram subsection collapsed into 1D. The profiles are in JSON format. The script and instructions for installation and use are available in a GitLab repository at https://gitlab.com/daviortega/ sideview-profile-average. The profiles were visualized with the ObservableHQ notebook located at https://beta.observablehq.com/@daviortega/generic-notebook-to-analyse-1d-averaged-electron-density-p. For each profile, we measured the distance between intensity peaks. These peaks correspond to the electron densities

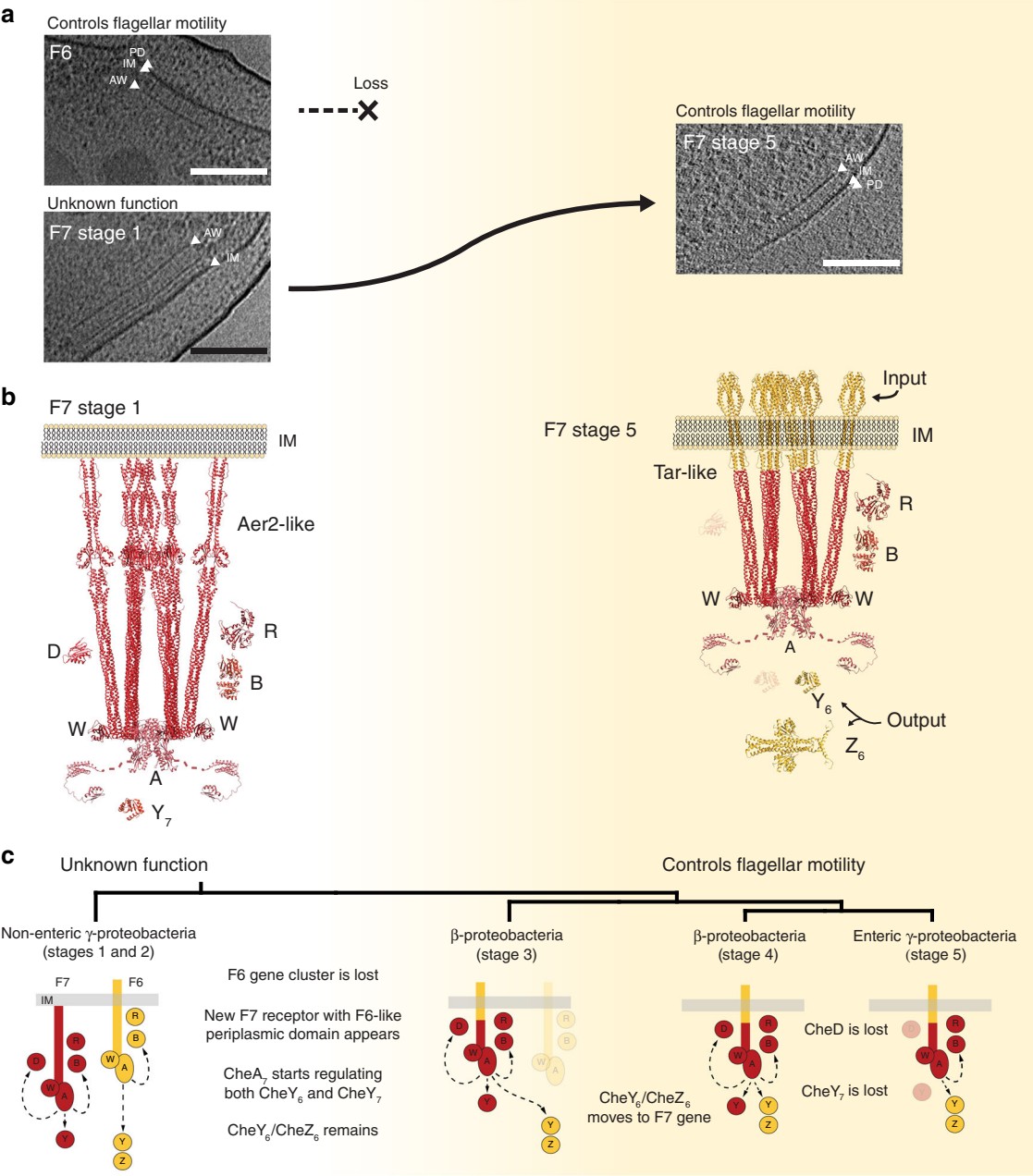

**Fig. 6 Evolution of the F7 chemosensory array in non-enteric γ-proteobacteria. F7 chemosensory systems acquire F6-like ultrastructure and function.**
**a** Tomographic slices showing F6 and F7 stage 1 chemosensory arrays in the same *P. aeruginosa* cell (left) and an F7 stage 5 chemosensory array in *E. coli* (right). Over the course of evolution, the F6 system is lost and the F7 system evolves similar ultrastructure and function to the F6 system. Features to identify chemoreceptors are highlighted: periplasmic domain (PD), inner membrane (IM), and CheA/CheW layer (AW). **b** Molecular models of F7 chemosensory arrays in *P. aeruginosa* (left) and *E. coli* (right) built based on[64,65]. Proteins displayed in this representation are: CheA (A), CheB(B), CheR (R), CheD(D), F7-CheY ($Y_7$), F6-like CheY ($Y_6$), and F6-like CheZ ($Z_6$). Models are colored according to their hypothetical original class: F7 (red) and F6 (yellow). **c** Working model of the evolution of the F7 chemosensory system in γ-proteobacteria and β-proteobacteria. Scale bars are 50 nm.

of the IM, the CheA/CheW baseplate and intermediate layers. Measurement uncertainty was estimated (coverage factor $k = 2$) for determining the center of each peak in pixels[44]. The measurements in Supplementary Table 4 were made using one array, and they agree with the values obtained from measurements in the sub-tomogram averages described above. The general distances reported for the layers in Fig. 2 are averages of the measurements in each organism with propagated uncertainty.

**Bioinformatics resources and software packages.** All sequences in this study were collected in the MiST database[45], domain architecture predictions from PFAM[46] were selected from SeqDepot[47], and 3D atomic models were taken from the Protein Data Bank (PDB)[48]. Domain architecture prediction was performed with CD-VIST (http://cdvist.zhulinlab.org/)[25]. We used CD-HIT v4.6 to reduce redundancy in unaligned sequences[49]. Multiple sequence alignments were performed with the algorithm L-INS-I from the package MAFFT v7.305b[50]. We used Gblocks v 0.91b[51] to eliminate poorly aligned columns in multiple sequence alignments. To perform sequence alignments with structural information we used STAMP from the MultiSeq[52] tool for VMD v1.9.3[53] which in turn was used to visualize and manipulate 3D structures. Homology modeling was performed using MODELLER v9.17[54]. Secondary structure predictions were performed with JNet Structure Predictions[55] in the Jalview v2.10.1[56] software, which was also used to visualize multiple sequence alignments. Similarity searches for sequences were conducted using BLAST v2.7.1+[57] and HMMER v3.1b2[58]. Phylogenetic reconstructions were performed using RAxML v8.2.10[59]. To collapse branches with low support in phylogenetic trees we used TreeCollapseCL4 v3.0[60]. Sequence logos were built using Weblogo 3.6.0[61]. Tomograms and model points were manipulated using 3dmod v4.9.9[40].

**Protein domain architecture prediction**. The domain architecture of the C-termini of chemoreceptors are poorly conserved among members of this protein family[62]. Further, protein domains commonly appearing in this region, such as HAMP[63] and PAS[26], are so diverse that in several instances predictive models have difficulty identifying them. To address this problem, we used CD-VIST on the two Aer2-like receptors without known domain architectures from *S. oneidensis* and *M. alcaliphilum* with TMHMM prediction, skipping HMMER3 and RPSBLAST steps, but adding three consecutive HHSEARCH steps against the PDB database with HHBLITS using uniclust30 at different thresholds for minimum probability: 60, 40, and 20%. Analyzing the CD-VIST domain coverage we predicted that *S. oneidensis*'s Aer2-like receptor has a PAS-PAS-HAMP-HAMP-MCPsignal domain architecture, similar to *V. cholerae* and that the *M. alcaliphilum*'s Aer2-like receptor has a HAMP-PAS-HAMP-HAMP-MCPsignal domain architecture. We further enhanced our confidence in these predictions by aligning the Aer2-like sequences to the sequence of the templates used to produce the homology models.

**Homology modeling**. To build homology models for the Aer2-like receptors in *V. cholerae*, *P. aeruginosa*, *S. oneidensis*, and *M. alcaliphilum* we used several crystal structures available in the Protein Data Bank (PDB), Supplementary Table 7. The files used in this process are described in Supplementary Table 8 and can be found in the Supplementary Data 1.

First, we built a homology model with two HAMPs followed by the MCPSignal domain that we name 2H + S. For that we used the structures 3ZX6 and the second HAMP of 4I3M to form a chimeric template. We manually aligned the structures of the templates against the Aer2 in *P. aeruginosa* (PA1076) and performed a multiple sequence alignment using L-INS-I and MultiSeq. To construct the homology model of this structure, we use MODELLER with the following parameters: a.library_schedule = autosched.slow, a.max_var_iterations = 1000, a.repeat_optimization = 100 and a.max_molpdf = 1e6. To make sure that the connection between both HAMPs remained the same, we added a restraint in both chains A and B from residues 359 to 385. We built 100 homology models with these parameters and chose the one with the lowest DOPE score.

Next, to add a PAS domain to this structure, we used the 3VOL and 4HI4 structures. First, we aligned chain B of 3VOL with the 2H + S homology model produced in the previous step. We noticed that this alignment produced clashes between the PAS domains. To overcome this obstacle, we used chains B and D in the 4HI4 structure as a model for the dimerization of the two PAS domains. We aligned chain B of 4HI4 to the 3VOL structure using the residues QWTDRT and then manually manipulated the dimer of PAS to be positioned in line with the 2H + S model to build the next homology model: P + 2H + S. Sequence alignment was performed as described before against the sequence of Aer2 in *P. aeruginosa* (PA1076). This homology model was used as the basis of the complete homology models of all the Aer2-like receptors.

To build the homology model of Aer2 in *P. aeruginosa* (PA1076), we used the P + 2H + S model together with the 4I3M structure. For that we manually aligned the structures to build the template. However, there is a 13 residues region unresolved in both structures (R156–G169) but predicted to be alpha helical. We assume that these two structures then are around 2.2 nm apart and took that into consideration while positioning the structures. Finally, the homology model was built using MODELLER with the parameters described above and with a restraint to force alpha helical conformation between residues 140 and 181.

To build the homology model of the Aer2-like receptor in *V. cholerae* (VCA1092) we used the P + 2H + S model together with 4HI4. The sequences of the templates and VCA1092 aligned pretty well with only a minor gap in the residues ELLRD, also predicted to be alpha helical. We aligned the end of chain B of the already aligned 4HI4 used in the P + 2H + S model to the beginning of chain A of P + 2H + P using STAMP and manually adjusted the position of the structures using VMD. The homology model was constructed with MODELLER and we imposed a restraint to force alpha helical conformation between residues 21–43 (C terminal) and 151–171 (unresolved gap).

To build the homology model of the Aer2-like receptor in *S. oneidensis* (SO_2123) we used the VCA1092 model since they have the same domain architecture. The sequences of the templates and SO_2123 also aligned pretty well with only a minor gap in the residues ESIDA, also predicted to be alpha helical. The C-terminal of the sequence is also predicted to be alpha helical up to the residue PHE7. We aligned the end of chain B of the already aligned 4HI4 used in the P + 2H + S model to the beginning of chain A of P + 2H + P using STAMP and manually adjusted the position the structures using VMD. The homology model was performed with MODELLER and we imposed a restraint to force alpha helical conformation between residues 21–43 (C terminal) and 151–171 (unresolved gap).

To build the homology model of the Aer2-like receptor in *M. alcaliphilum* (MEALZ_2872) we used the P + 2H + S model and the 4I3M structure. To find out which of the 3 HAMPs in the 4I3M structure is most closely related to the C-terminal HAMP of MEALZ_2872 we used BLAST to find HAMP sequences in the *Pseudomonas* group similar to each of the HAMPs in the 4I3M and to the C-terminal HAMP of MEALZ_2872. We aligned the sequences using L-INS-I and perform a phylogenetic reconstruction using RAxML with -m PROTGAMMAIA UTO -p 1234555 -x 9876545 -f a -N 100 as parameters. Tree nodes were collapsed to a certainty score of 50. The phylogenetic analysis showed that the C-terminal HAMP of MEALZ_2872 is closely related to the second HAMP of 4I3M. We

truncated the 4IM3 structure to contain only the second HAMP and aligned an extended helix connecting to the third HAMP with the PAS domain of the P + 2H + S model. We used this alignment to place the HAMP at the right position and deleted the extended helix. These structures were used as a template for the MEALZ_2872 homology model built with MODELLER as described above and with restraints to force alpha helical conformation in residues 194–216, 255–270, and 721–728.

**Chemotaxis system classification**. Relevant protein sequences of chemotaxis components were classified using HMMER and the hidden Markov models previously published[9]. The model with highest score was used to assign chemotaxis components to classes.

**F7 system identification in γ-proteobacteria**. To estimate how widespread F7 systems are in γ-proteobacteria, we randomly picked 310 genomes from γ-proteobacteria from MiST. From those, we selected the CheA protein sequences and then classified them using HMMs provided by the authors of ref. [9]. The CheA proteins classified as F7 systems belonged to 176 genomes. Supplementary Table 9 lists all 310 genomes and marks the presence of the F7 systems in the 176 genomes.

**Phylogenetic tree of F7 systems in proteobacteria**. To build a tree of the F7 and F8 systems in proteobacteria we used a concatenated alignment of the protein sequences of CheA, CheB, and CheR, as previously described[9]. We first collected every CheA belonging to these two classes from 1152 proteobacteria genomes in MiST (547 from F7 class and 168 from F8 class) and used CD-HIT to eliminate redundancy at the 85% identity level (201 from F7 class and 119 from F8 class). To find CheB and CheR proteins that confidently function with the selected CheAs, we searched for genes that code for these proteins in the range of ten genes upstream and downstream from each *cheA* gene. Conflicts of multiple or missing *cheB*, *cheR*, or *cheA* genes within that range were manually resolved or the system was removed from the dataset. At this stage the dataset contained 272 protein sequences of CheAs, CheBs, and CheRs. We aligned each protein individually with L-INS-I from MAFFT. We used Jalview to examine the alignment and removed ten sequences for not being complete genes and realigned the sequences with L-INS-I. The final dataset had 262 sequences from F7 (168) and F8 (94) systems from 246 genomes. For each protein family, we used Gblocks to remove alignment positions with low information. The Gblock parameters were: b3 = 8 -b4 = 10 b5 = h. The resulting alignments of the protein sequences of CheA, CheB and CheR were concatenated into a single alignment with 698 columns. We used RAxML with parameters -m PROTGAMMAIAUTO -f d -d -N 25 with different seeds 10 times and 3 partitions set to evolutionary model AUTO with boundaries 1–312, 313–559, and 560–698 to accommodate possible differences in the evolutionary models selected for CheA, CheB, and CheR sequences. We selected the tree with best maximum likelihood score. We also ran 1000 rapid bootstrap on the same alignment with the parameters -m PROTGAMMAIAUTO -p 1234555 -x 9876545 -f a -N 1000. We mapped these bootstrap values to the best tree and used TreeCollapseCL4 to collapse nodes with less than 50% uncertainty to polytomies. To find an appropriate rooting point, we built an auxiliary phylogenetic inference that included sequences from the ancient class F1 as an outgroup (Supplementary Fig. 1—inlet). Following the exact same protocol, we also generated an auxiliary phylogeny that contained the same sequences and included the CheABR of three systems from the class F1 (*B. subtilis*, *Thermatoga maritima*, and *Clostridium thermocellum*), Supplementary Fig. 1—inlet. Although our analysis was unable to unambiguously determine the exact placement of the last common ancestor of the classes F7 and F8, it showed a polytomy more ancient than all F7 in both γ- and β-proteobacteria. We chose to root the CheABR phylogeny separating the F7 present in most α, γ, and β-proteobacteria from the rest of the sequences from the F7 and F8 classes, coherent with the CheABR analysis of Wuichet and Zhulin[9]. We also mapped the CheA gene neighborhoods (15 genes up and downstream) to the CheABR tree using custom scripts written in Python to produce Supplementary Fig. 1. BLAST all vs. all to all CheAs and selected neighboring genes was used to loosely define homologous sets of proteins with at least 10E−40 E-value and query coverage of 95% to any member of the set. As an exception to this rule, the anti-sigma factor antagonists were selected with the threshold of 1E−5 and query coverage of 50%. Homologs of relevant proteins are highlighted in different colors. We manually selected representatives of relevant genes neighboring CheA for major branches relevant to this study for display in Fig. 3. The phylogenetic trees can be found in Supplementary Data 2.

**Phylogenetic profiles of F6 and F7 systems**. We first selected the genomes of the organisms we imaged: *M. alcaliphilum*, 20Z, *P. aeruginosa* PAO1, *S. oneidensis* MR-1, *V. cholerae* O1 biovar El Tor str. N1696. In order to perform phylogenetic profiling of the chemotaxis systems in γ-proteobacteria, we added 162 genomes from this class and 10 genomes from β-Proteobacteria as an outgroup, for a total of 176 genomes (Supplementary Table 10). The number of selected genomes is coincidently the same as the number of genomes from γ-proteobacteria with F7 systems described above but only a fraction of genomes are present in both sets. To build the organism tree in Supplementary Fig. 2 we used the same procedure as described in ref. [27] with the difference that the final concatenated alignment served

as an input to RAxML to generate 164 inferences with the parameters -m PROTGAMMAIAUTO -p 12345 -f d -d -N 164. Chemotaxis proteins from these genomes were classified as described above and mapped onto the organism tree to produce Supplementary Fig. 2. The phylogenetic trees can be found in Supplementary Data 2.

**Domain architecture prediction**. We selected the protein sequences of chemoreceptors present in the gene neighborhood used to build Supplementary Fig. 1 and use CDVIST to predict the domain architecture using TMHMM, HMMER3 against Pfam 30.0 database. The results are shown in Supplementary Fig. 3.

**Identification of Aer2-like and McpA-like receptors**. We first collected all 3389 chemoreceptors from the 176 genomes used to build the phylogenetic profiles. We defined a protein as a chemoreceptor if it contained the MCPsignal PFAM domain. Then we grouped them in clusters of orthologous groups using the same technique described in ref. [18]. To pick Aer2-like receptors we used an $E$-value of $1E-135$ and selected all 144 receptors present in the same group as the Aer2 (PA1076) from *P. aeruginosa*. From those, we removed six receptors from the β-proteobacteria outgroup, five that were not classified as 36H receptors, and three other sequences that did not seem to align well with the group. The final set of Aer2-like receptors had 130 Aer2-like receptors and was aligned using L-INS-I and manually inspected with Jalview. Chemoreceptor families and subfamilies are prone to have diverse C-terminal domain architectures so following the procedure in ref. [62], we manually trimmed the sequences to only contain the regions common to all receptors. This final alignment was used to build a phylogenetic tree with RAxML. We built 200 independent inferences with parameters -m PROTGAMMAILG -p 1234555 -f d -d -N 200 and 1000 rapid bootstrap trees with -m PROTGAMMAILG -p 1234555 -x 9876545 -f a -N 1000. Bootstrap scores were mapped to the tree with best maximum likelihood from the 200 independent inferences. Nodes were collapsed to polytomies at 50% uncertainty using TreeCollapseCL4. The same procedure was executed to make the tree of McpA-like receptors but with an E-value threshold of $1E-30$. The McpA-like cluster was defined as the one containing McpA from *P. aeruginosa* (PA0180). There were 40 McpAs in the final dataset. Both trees are displayed in Supplementary Fig. 5.

**Identification of pentapeptide in receptors**. We used regular expressions to find the -x-[HFYW]-x(2)-[HFYW]- motif in the sequences of Aer2 and McpA used to build Supplementary Fig. 4. Supplementary Table 11 shows the pentapeptides for Aer2. Because no McpA sequence matched this motif but it had a conserved C-terminal domain, we built a sequence logo for the conserved region, Supplementary Fig. 4. We also show the conserved C-terminal for each McpAs in Supplementary Table 12.

**Phylogenetic tree of CheY**. The CheY protein comprises a single domain, known in the PFAM database as Response Regulator (Response_reg). However, this domain appears in several other proteins as well. In order to select proteins with one and only one response regulator domain, we collected the domain architecture information from PFAM v30 and predicted transmembrane regions by TMHMM from SeqDepot for all sequences from the 246 genomes and used Regular Architecture (https://www.npmjs.com/package/regarch) to filter only single CheY domains with a specific rule, Supplementary Note 1. This pattern selected 4941 sequences. We then used the same clustering techniques described for Aer2-like and McpA-like receptors with an E-value threshold of $10E-30$ and selected the largest group, with 1394 sequences. This group contains the known CheYs of the model organisms in this study and others. We aligned this dataset with L-INS-I and manually removed 3 sequences that were highly divergent using Jalview. We built the tree with 500 rapid bootstraps with RAxML and searched for the best tree of this set with the parameters -m PROTGAMMALG -p 1234555 -x 9876545 -f a -N 500. Finally, we collapsed nodes with less than 50% support into polytomies using TreeCollapseCL4 to produce Fig. 4.

**Analysis of CheY sequences**. There is no model to classify CheY proteins into chemosensory classes. To find CheY-F6-like sequences in F7 containing genomes we used CheZ as a marker to find CheY. For each stage (3–5), we selected the genomes from the tree in Supplementary Figure 2 and selected CheZ-F7 from our previously classified dataset of CheZ protein sequences. Next, we used MiST3 to select one gene upstream and one downstream of each cheZ and search if the gene is present in the branch containing CheY's known to participate in flagellar control in Fig. 4. In the case of the CheY-F6 we used the same protocol, but starting the search with all genomes in Supplementary Table 9. We selected 64 CheY-F6, 22 CheY-stage3, 27 CheY-stage4, and 24 CheY-stage5. We eliminated two sequences from CheY-F6 that opened a minor gap for the sake of clarity without changing the results of the sequence logo. We used Weblogo to build the sequence logos in Fig. 5A. Positions that were conserved in all groups but the same in stage 3–5, and different in CheY-F6 were mapped in the CheY NMR 3D model structure 2LP4[32] using VMD, Fig. 5B.

**Reporting summary**. Further information on research design is available in the Nature Research Reporting Summary linked to this article.

## Data availability

Tomograms are available in the Electron Tomography Database—Caltech at https://etdb.caltech.edu and their identifiers are listed in the Supplementary Table 6. Phylogenetic trees in Fig. 4, Supplementary Figs. 1, 2, and 4 are available in Supplementary Data 1. The homology models in Fig. 2D are available in Supplementary Data 2. Other data are available from the corresponding authors upon reasonable request.

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

## Acknowledgements

The authors wish to thank Drs. Zhiheng Yu, Jason de la Cruz, Chuan Hong, and Rick Huang for microscopy support at HHMI Janelia Farms; Dr. Mohamed Y. El-Naggar for insights into the culturing of *S. oneidensis*, Dr. Kristin Wuichet for discussions about the evolutionary interpretation of the bioinformatics data and Dr. Keith Cassidy for discussions on homology model building and for providing the Tar model used in Fig. 5. We also thank Dr. Catherine M. Oikonomou for helpful discussion and suggestions on the paper. This work was supported by the John Templeton Foundation as part of the Boundaries of Life Initiative (grants 51250 and 60973), NIGMS grant R35 GM122588 (to G.J.J.), NIGMS grant R01 GM108655 (to K.J.W.), the Max-Planck-Gesellschaft (to S.R.), the National Science Foundation grant CBET-1605031 (to M.G.K.), the Swiss National Science Foundation (R.K.), and the Air Force Office of Scientific Research Presidential Early Career Award for Scientists and Engineers grant FA955014-1-0294 (S.P.). *P. aeruginosa* strains were acquired from the transposon mutant collection that was made possible by NIH grant P30 DK089507.

## Author contributions

Conceptualization: D.R.O., A.B., and G.J.J.; Formal analysis: D.R.O.; Funding acquisition: K.J.W., M.G.K., S.R., A.B., and G.J.J.; Investigation: D.R.O., P.S., P.M., A.K., S.C., K.J.W., S.P., S.A.C., R.K., and A.B.; Methodology: D.R.O., W.Y., P.S., P.M., K.J.W., S.P., D.A.C., M.G.K., S.R., and A.B. Project administration: G.J.J.; Software: D.R.O.; Supervision: M.G.K., S.R., A.B., and G.J.J. Validation: D.R.O., W.Y., P.S., P.M., A.K., S.C., K.J.W., S.P., S.A.C., R.K., and A.B.; Visualization: D.R.O., W.Y., P.S., S.P., and A.B.; Writing—original draft: D.R.O., P.S., A.B., and G.J.J. Writing—review and editing: D.R.O., W.Y., P.S., K.J.W., S.P., D.A.C., M.G.K., S.R., A.B., and G.J.J.

## Competing interests

The authors declare no competing interests.
