## [Peer Review File · Nature Communications]

Reviewers' comments:

Reviewer #1 (Remarks to the Author):

This study by Ortega et al. explores the evolutionary diversity and origin of bacterial chemosensory systems. It combines bioinformatic analysis with structural observations of chemotactic arrays from electron cryotomography of four bacterial species. The authors observed a novel tall array architecture in tomograms of these four species, and then used the absence of these tall arrays in mutant stains of *Vibrio cholerae* and *Pseudomonas aeruginosa* to assign the identity of the tall arrays to the F7 system. Juxtaposing "1D electron density profiles" of the arrays next to Aer2 homology models, the authors conclude that the F7 arrays are constructed from Aer2-like receptor proteins. Finally, through phylogenetic analysis of proteobacterial genomes, the authors propose that the ancient F7 chemosensory system incorporated the F6 flagellar control system to produce the modern F7 system present in *Escherichia coli*.

I do not have the background to evaluate the bioinformatic analysis in this manuscript, but it appears to be quite thorough. As for the tomographic structural analysis of F7 arrays, I very much appreciate the approach of comparing changes in macromolecular architecture to reveal evolutionary relationships. However, the minimal structural analysis of these arrays needs to be improved in order to support the authors' conclusions.

In Figure 2, the authors make "1D electron density profiles" from the tomograms and then compare these profiles to homology models of Aer2-like receptor proteins from the four species. As far as I can tell from the methods section, the density profiles are simple intensity line scans through one 2D tomographic slice of a single chemotactic array (I think the line width used for averaging the profile is not specified in the methods). This is N=1: one cell and only one measurement through a single 2D slice of that cell. This is not sufficient analysis to make strong structural conclusions about the chemotactic arrays.

Ideally, 3D subtomogram averaging should be performed using multiple subtomograms (at least 100) from multiple cells (at least 4). I appreciate that some regions of the chemotactic arrays may be flexible and therefore poorly resolved in the average. However, if the spacing of the major layer densities is generalisable between arrays of a given species, as the authors claim, then these structural features will be seen in the average. If this 3D analysis proves too challenging, at the very least 2D averaging should be performed, again using at least 100 2D tomographic slices from at least 4 cells. If the layer pattern of the major densities is not resolved in such an average, then it is not a generalisable structural feature.

Furthermore, the "1D electron density profiles" seem to not match the homology models very well. There do not appear to be globular domains in the homology models of any of the four species that match the L1 layer. The *P. aeruginosa* homology model also lacks a domain to account for the L3 layer. Perhaps I am misunderstanding and the Aer2 homology models are not supposed to account for all the layers of cytosolic density seen in the arrays? Are additional proteins expected to be assembled into the arrays? If so, this was not clear to me from reading the manuscript.

As it is, I do not believe that the structural analysis contributes enough to the paper. However, if the authors employ 3D or 2D averaging that shows reproducible species-specific differences in array structure, then I believe the paper can be strongly considered for publication.

Minor points:

- In Figure 1, it would be helpful to readers if the authors could diagram the architectural features that distinguish F6, F7, and F9 arrays. To an untrained eye, F6 and F7 look quite similar.

- The labeling and figure legend of Figure 2 do not indicate that the structures being compared are all F7 arrays. This should be clearly noted.
- The legend of Figure 2 should explain the abbreviations IM, L3, L2, L1, SL, AW, H, P, S. Similarly, the legend of Figure 5 should explain the abbreviations PD, IM, AW, D, W, A B, R, Y7, Y6, Z6.
- The "1D electron density profiles" in Figure 2 lack a labeled X-axis.
- Line 499: By "Gatan imaging filters" do the authors mean "Gatan energy filters"?
- Line 501: $e^{-}/\text{\AA}^2$ should be $e^{-}/\text{\AA}^2$.
- Also in the electron cryotomography section of the methods, what was the magnification / pixel size used for acquisition?

Reviewer #2 (Remarks to the Author):

This manuscript uses bioinformatic analysis, in conjunction with information from cryo-electron tomography, to trace the evolution of the chemotaxis system of *Escherichia coli* and related organisms. The principle findings are that this system, the paradigm for our understanding of bacterial chemosensory systems, is a hybrid of two more ancient systems and that its transmembrane chemoreceptors are chimeras generated by gene fusion. These are novel and fascinating findings that will be of interest to most involved in the study of bacterial chemotaxis and, more widely, to many involved in the study of bacterial sensory signaling. The manuscript is lucidly written. In large part, the distinction between observation and interpretation is clearly defined. However, I have one major concern and some additional comments and suggestions.

Major concern.

Multiple density layers. The presence of the multiple density layers besides the AW layer is not convincingly documented for the chemosensory arrays of the four species shown in Fig. 2A. At least to this reader's eyes, for *V. cholerae* none of the L layers is clearly shown in the image provided. The SL layer is obvious only for *M. alcaliphilum*, not for the other three species. For *S. oneidensis*, identification of L1 and L3 layers is unconvincing. Given these issues, the correlation the authors suggest (Fig. 2B and the relevant text in lines 171-181, 380-395) between the layers and structural features of the four receptor species would not convince a critical reader. The authors need to find a compelling way to illustrate the presence of the layers and the correlation with receptor structural features that they suggest.

Other comments and suggestions.

1. Genes versus protein products. Throughout much of the text, the authors do a good job correctly distinguishing the two and following the convention of gene names in lower case italics; protein names in regular font with the first letter capitalized. There are a few instances where this is not the case.
 - a. Line 251. Insert "gene" after "chemoreceptor".
 - b. Line 253. *tar*, *tap*, *trg* and *tsr* should be *Tar*, *Tap*, *Trg* and *Tsr*.
 - c. Lines 280-282. The several gene names in these lines are used to modify "receptors", thus protein names should be used or "receptors" should be changed to "receptor genes".
2. Lines 45-47. This compound sentence is awkward. It would read better as two separate sentences.
3. Line 168. Concluding from four examples that tall F7 arrays are "widespread" seems an overstatement. A revised statement would be better.
4. Line 372. I suggest using a word other than "patches" since it has been used to refer to

chemoreceptor/chemosensory arrays and thus could cause confusion.

5. Lines 433-457. In this text PCR primer sequences are written with lower case letters. Shouldn't upper case letters be used?

6. Fig. 1. The "empty" arrows are difficult to distinguish from the black arrows when this figure is printed on a page as a hard copy. Thus it would be best to change the empty arrows to something easier to distinguish on a printed copy from the black arrows.

Reviewer #3 (Remarks to the Author):

Previous work by the Zhulin group has shown that chemosensory pathways can be classified into flagellar systems F1-F17, Type IV pili associated functions and alternative functions. Of these systems, class F6 and F7 are particularly relevant since present in many species. Whereas F6 systems control flagellar rotation, the function of F7 systems are less well understood. Whereas the chemotaxis system of *E. coli* belongs to F7, the F7 system of *P. aeruginosa* is not involved in chemotaxis and of unknown function. The authors have conducted studies to understand the co-existence of F6 and F7 systems that differ in function.

Using electron cryotomography the authors observe two distinct chemoreceptor arrays in a number of species that differ in their width. Using different mutants the authors establish that the thinner arrays are F6 systems, whereas F7 systems form thicker arrays. Whereas canonical F6 systems contain transmembrane receptors with periplasmic/extracytosolic sensor domains the authors show that the F7 visualised by cryo-EM use soluble receptors that in many cases lack any apparent transmembrane regions. In those arrays, next to the densities caused by bound cheA/CheW, the authors observe additional layers, L1 to L3.

In the second part of this manuscript the authors establish the evolutionary history of these systems. They show that F6 and F7 systems existed contemporaneously. However, they were able to distinguish 5 different steps in the evolution of the F7 system. A key event in this evolutionary process is the recruitment of the CheY protein from an F6 pathway permitting that the output occurs at the level of the flagellar motor. Data indicate that the F7 pathways are taking over and that F6 pathways will eventually get lost. The F7 system of *P. aeruginosa* corresponds thus to an ancient systems (stage 2 in the evolution).

This manuscript is very well organized and written. The combination of cryo-EM with bioinformatics was shown to be very powerful to address a very complex issue. This manuscript provides solid answers to a number of puzzling issues in this field such as for example why some F7 systems carry out an F6 functions whereas other F7 systems have clearly a function that is different. Many bacteria have multiple chemosensory pathways and this work will permit to get initial information of these pathways.

This referee has several comments and propositions on how to improve this manuscript.

1. A significant part of literature on the *P. aeruginosa* Aer2 chemoreceptor uses with McpB a synonym. Both terms are not very fortunate since this receptor is unlikely to be an aerotaxis receptor nor a methyl-accepting chemotaxis protein. However, for clarity it should be mentioned that there is a synonym. One issue has not been mentioned in this study and its assessment may provide further information into Aer2/McpB type chemoreceptors. Of the 26 *P. aeruginosa* chemoreceptors only two possess a C-terminal extensions with a terminal pentapeptide for CheR/CheB binding, which are McpB and McpA (Fig. 1 in PMID:24714571). Interestingly, both of receptors are encoded in the F7 gene cluster. In the case of McpB, this terminal pentapeptide is of crucial importance: it binds with nanomolar affinity to CheR2 and no other CheR paralogue, removal of this pentapeptides prevents it interaction with Aer2/McpB and abolishes receptor methylation. Interestingly the terminal pentapeptide of McpA appears to be some sort of degenerated: whereas the McpB pentapeptide matches the consensus defined by the Stock laboratory X-W-X-X-F, the McpA pentapeptide has the W replaced by V. In contrast to McpB, the McpA pentapeptide did not bind to any of the four CheR. There are several questions: Is the possession of functional C-terminal pentapeptides a general feature for the family of Aer2-like

receptors? Is this a feature common to receptors that feed into step 1 and step 3 F7 pathways. Do all McpA possess a terminal pentapeptide? Is there evidence that ancestral McpA had a functional pentapeptide that during evolution has lost its capacity to bind to CheR?

In addition, a very interesting observation is the existence of additional density layers in these F7 systems, L1 to L3. In *E. coli* CheR binds with much higher affinity to the C-terminal pentapeptide of Tar and Tsr than to receptor variants from which the pentapeptide had been removed (binding to the methylation site). In *P. aeruginosa* removal of the pentapeptide prevented CheR binding in vitro. The L1 layer corresponds to the zone of the C-terminal pentapeptide. Could this layer represent bound CheR/CheB? Have the authors analysed an Aer2 mutant from which the pentapeptide had been removed? The authors argue that CheR is present at low concentration. However, the Hazelbauer group has shown that CheB of *E. coli* also binds to this pentapeptide, although with lower affinity than CheR. Phosphorylation of CheB was shown to enhance the binding affinity.

2. The key event in this evolutionary process is the recruitment of the F6 CheY into the F7 pathway. This incorporation needs to be accompanied by mutations in CheY enabling it to interact with the F7 CheA. Can sequence analyses provide any clues as to the changes in the F6 CheY?

3. In stage 4 pathways there are two CheY. Has the literature on these systems been inspected to get any evidence supporting the notion that the newly incorporated CheY plays a role in chemotaxis?

4. Both, the F6 and stage 3 -5 F7 systems mediate chemotaxis and the authors conclude that the F7 systems appear to take over. To enhance the clarity could the gene order of a typical F6 system be added to Fig. 3? In the Discussion maybe the authors want to compare both systems and hypothesize on their potential functional advantages/disadvantages in mediating chemotaxis?

Reviewer #4 (Remarks to the Author):

This manuscript describes an unusual and interesting effort to combine cryo-electron imaging, bioinformatics and evolutionary biology to describe the trajectory of a key family of bacterial sensory systems. Such cross-disciplinary efforts have the potential to make breakthroughs in problems that have proven too difficult to address by a single approach.

While the manuscript's conclusions are generally convincing, this reader was unable to understand how the gene phylogenies were completely parallel with organismal phylogeny. Perhaps the authors simply need to make a clearer explanation to the reader that addresses the apparent inconsistencies noted in this review.

A copy of the manuscript is attached, with comments in the tracked-changes.

Ned Ruby

We have addressed in detail all the reviewers' comments below. The changes in the main text are highlighted in blue.

Reviewer #1 (Remarks to the Author):

This study by Ortega et al. explores the evolutionary diversity and origin of bacterial chemosensory systems. It combines bioinformatic analysis with structural observations of chemotactic arrays from electron cryotomography of four bacterial species. The authors observed a novel tall array architecture in tomograms of these four species, and then used the absence of these tall arrays in mutant stains of *Vibrio cholerae* and *Pseudomonas aeruginosa* to assign the identity of the tall arrays to the F7 system. Juxtaposing “1D electron density profiles” of the arrays next to Aer2 homology models, the authors conclude that the F7 arrays are constructed from Aer2-like receptor proteins. Finally, through phylogenetic analysis of proteobacterial genomes, the authors propose that the ancient F7 chemosensory system incorporated the F6 flagellar control system to produce the modern F7 system present in *Escherichia coli*.

I do not have the background to evaluate the bioinformatic analysis in this manuscript, but it appears to be quite thorough. As for the tomographic structural analysis of F7 arrays, I very much appreciate the approach of comparing changes in macromolecular architecture to reveal evolutionary relationships. However, the minimal structural analysis of these arrays needs to be improved in order to support the authors' conclusions.

In Figure 2, the authors make “1D electron density profiles” from the tomograms and then compare these profiles to homology models of Aer2-like receptor proteins from the four species. As far as I can tell from the methods section, the density profiles are simple intensity line scans through one 2D tomographic slice of a single chemotactic array (I think the line width used for averaging the profile is not specified in the methods). This is N=1: one cell and only one measurement through a single 2D slice of that cell. This is not sufficient analysis to make strong structural conclusions about the chemotactic arrays.

Ideally, 3D subtomogram averaging should be performed using multiple subtomograms (at least 100) from multiple cells (at least 4). I appreciate that some regions of the chemotactic arrays may be flexible and therefore poorly resolved in the average. However, if the spacing of the major layer densities is generalisable between arrays of a given species, as the authors claim, then these structural features will be seen in the average. If this 3D analysis proves too challenging, at the very least 2D averaging should be performed, again using at least 100 2D tomographic slices from at least 4 cells. If the layer pattern of the major densities is not resolved in such an average, then it is not a generalisable structural feature.

We appreciate the reviewer's concern that we only used a single cell to generate the 1D profile. For clarification, the 1D profile was generated using an average of a stack of 2D slices through the 3D volume. Therefore, the profile is an accurate measurement of the chemoreceptor arrays in 3D. To further support our conclusions, we performed 3D sub-tomogram averages with more cells will improve the manuscript as suggested by the reviewer.

We carried out 3D sub-tomogram averaging of the F7 arrays from multiple cells for each strain used for structure analysis. The numbers of the cells and the sub-tomograms used to generate averages are in the table below (now Table S3).

	Cells (tomograms)	Sub-tomograms	Pixel size (nm)
P. aeruginosa	5	1113	0.64
V. cholerae	6	265	1.3
S. oneidensis	5	327	1
M. alcaliphilum	2	1448	1

1. The percentage of cells that contain F7 arrays is low, particularly for *M. alcaliphilum*. Since there is no method available to increase expression of this system, we have limited datasets that are suitable for sub-volume averaging. However, the sub-tomogram average of *M. alcaliphilum* based on the arrays found in two cells possesses the expected features based on the bioinformatics analysis.
2. Since our aim was to visualize complete cells, we chose a large pixel size for data collection. Combined with the inherent thickness of the bacterial cells, the resolution of the datasets is limited.
3. Last but not least, the size of the F7 arrays varies significantly among different bacterial strains. For example, in *V. cholerae*, the F7 arrays are smaller compared to the F6 arrays (see figure below). This limits the numbers of available sub-volumes.

The density variance along the receptor in the F7 arrays agrees with the results from the 1D electron density profiles. All of the resolved features are in accordance with the previously reported results from the 1D profile averages.

Additionally, we confirmed the conserved 12 nm hexagonal packing of the receptors that are common to all chemotaxis arrays imaged so far.

Based on these new results, we adapted the text accordingly:

Starting from line 175:

“To better visualize these additional layers, we computed the average 1D profile of electron density from the CheA/CheW layer to the IM in each species (Fig. 2A).”

We changed to (now line 183):

“To better visualize these additional layers, we computed 1D profiles as well as 3D sub-tomogram averages of the F7 array in each species (Fig. 2, Table S3).”

Also, we removed the layer SL since they are not very clear in the sub-tomogram average, and their designation remains unclear. For that reason, we adapted the following sentence in Line 177:

“Starting from the CheA/CheW layer and moving toward the membrane, *P. aeruginosa*, *S. oneidensis*, and *M. alcaliphilum* showed a density layer very close to the CheA/CheW layer, which we refer to as signaling layer SL. All four species then exhibited two higher layers we name L1 and L2. Finally, all but *M. alcaliphilum* presented an additional layer, L3, near the IM.”

To (now line 185):

“All strains contain additional density layers between the CheA/CheW layer and the inner membrane. We labeled them in order from the CheA/CheW layer towards the inner membrane as L1, L2 in *M. alcaliphilum* and L1, L2, and L3 in *P. aeruginosa*, *V. cholerae* and *S. oneidensis* (Fig. 2C). Some of these layers consistently correlate with the PAS domains of the receptors in each species (Fig. 2D).”

Below is the new Figure 2 and legend:

Fig. 2: Structural analysis of F7 chemotaxis arrays in *P. aeruginosa*, *V. cholerae*, *S. oneidensis*, and *M. alcaliphilum*. A) Side-view of F7 chemotaxis arrays (scale bar: 50 nm) relative to the inner membrane (IM) and outer membrane (OM). B) Top-view of a sub-tomogram average at the CheA/CheW layer reveals that F7 arrays have the typical hexagonal packing of receptors with ~12 nm spacing (scale bar: 20nm). C) Side view of the sub-tomogram averages. Several density layers (L1, L2, and L3) are present between the inner membrane and the CheA/CheW layer (AW) (scale bar: 20nm). D) A comparison of the side-views from the sub-tomogram averages with homology models of chemoreceptors (to scale). The presence of PAS domains in the receptor structures consistently correlate with L2 layer in all organisms and the L3 layer in *V. cholerae* and *S. oneidensis*. The box diagram shows the PFAM protein domains of each receptor: PAS (P), HAMP (H), and MCPsignal (S).

We changed the CryoEM methods to the following:

“Electron cryotomography

Cells were prepared for electron cryotomography as described previously⁵³. Images were collected using either an FEI G2 300 keV field emission gun microscope or an FEI TITAN Krios 300 keV field emission gun microscope with lens aberration correction (FEI Hillsboro, OR). Both microscopes were equipped with Gatan energy filters and ‘K2 summit’ counting electron detector cameras (Gatan, Pleasanton, CA). The data collection software used to collect the tilt series was UCSFtomo⁵⁴. The cumulative electron dose was 160 e⁻/Å² or less for each individual tilt series. The tomograms used in this study are available in the Electron Tomography Database – Caltech⁵⁵ and their identifiers can be found in Table S6.

Image analysis

CTF correction, frame alignment and SIRT reconstruction was done using the IMOD software package^{56,57}. Sub-tomogram averaging was carried out with the Dynamo software package^{58,59}. F7 arrays were modeled as patches of surfaces in individual tomograms, and then particles containing inner membrane and baseplate were cropped out based on the geometry of the surface model. Information about the number of cells, particles and pixel sizes are summarized in Table S3. Particles were aligned first based on the inner membrane and baseplate density, and then subsequently aligned with in-plane rotations and shifts. 2D tomoslices of the averages represent the top view and the side view of the repeating unit of the F7 arrays. A side view image was used for comparison to the receptor homology models.”

We also changed the methods section “1D electron density profiles” to “Distance measurements of electron density layers” and edited the explanation to clarify to the readers how the algorithm builds the profiles:

“Distance measurements of electron density layers

To measure the distance between the inner membrane (IM) and the electron density layers, we used a custom script written in Node.js. The tool is available on node package manager (npm): <https://www.npmjs.com/package/sideview-profile-average>. The script uses a tomogram and model points that follow the inner membrane at a given 2D slice as input, Figure S6. The final average profile is representative of a 3D tomogram subsection collapsed into 1D. The profiles are in JSON format. The script and instructions for installation and use are available in a GitLab repository at <https://gitlab.com/daviortega/sideview-profile-average>. The profiles were visualized with the ObservableHQ notebook located at <https://beta.observablehq.com/@daviortega/generic-notebook-to-analyse-1d-averaged-electron-density-p>. For each profile, we measured the distance between intensity peaks. These peaks correspond to the electron densities of the IM, the CheA/CheW baseplate and intermediate layers. Measurement uncertainty was estimated (coverage factor k = 2) for determining the center of each peak in pixels⁶¹. The measurements in Table S4 were made using one array, and they agree with the values obtained from measurements in the sub-tomogram averages described above. The general distances reported for the layers in Figure 2 are averages of the measurements in each organism with propagated uncertainty.”

The new Figure S6 and its legend are as follows:

Fig S6: The 1D electron density profile is a collapse of a 3D sub-volume. For each model point (red), the algorithm extends a profile perpendicular to the model points (blue). Then, to calculate the intensity of each pixel in the profile, it averages the intensity of the pixels perpendicular to the profile in the slices above and below (green). The final 1D profile is an average of the profiles calculated for each pixel of the model point. Effectively, this is a 1D collapse of a 3D sub-volume.

Furthermore, the “1D electron density profiles” seem to not match the homology models very well. There do not appear to be globular domains in the homology models of any of the four species that match the L1 layer. The *P. aeruginosa* homology model also lacks a domain to account for the L3 layer. Perhaps I am misunderstanding and the Aer2 homology models are not supposed to account for all the layers of cytosolic density seen in the arrays? Are additional proteins expected to be assembled into the arrays? If so, this was not clear to me from reading the manuscript.

At present, we are only confident that the PAS domains in all four species correlate with the density layers. More specifically, the density layers match all PAS domains in 6 occurrences: 1 in *P. aeruginosa*, 2 in *V. cholerae* and *S. oneidensis*, and 1 in *M. alcaliphilum*.

However, we were also puzzled by the L1 layer in all organisms and the L3 layer in *P. aeruginosa*. We addressed this in the original manuscript in the discussion section line 387:

“The L1 layer matched the junction between the HAMP and signaling domains, which is puzzling because this chemoreceptor region is predicted to have low molecular density³⁵. It is

unlikely that this density is produced by another known chemotaxis protein. For example CheR, that binds the chemoreceptor in that area, is not expected to have enough abundance to generate a visible density layer^{36,37}. Furthermore, previous cryo-ET of *in vitro* preparations containing only *E. coli* CheA, CheW and Tsr showed a similar layer in that region, suggesting one or more of these proteins alone is responsible for the L1 layer³⁸.”

Even though it is unclear what composes these layers, we think that it is worth mentioning the consistent position of these layers at a specific receptor height from the AW layer.

We changed the text in the original manuscript to (now line 471):

“The L1 layer matched the junction between the HAMP and signaling domains, which is puzzling because this chemoreceptor region is predicted to have low molecular density³⁵. It is unlikely that this density is produced by another known chemotaxis protein. For example CheR, that binds the chemoreceptor in that area, is not expected to have enough abundance to generate a visible density layer^{36,37}. Furthermore, previous cryo-ET of *in vitro* preparations containing only *E. coli* CheA, CheW and Tsr showed a similar layer in that region, suggesting one or more of these proteins alone is responsible for the L1 layer³⁸.”

Additionally, we added:

“Similarly, the L3 layer in *P. aeruginosa* F7 array appears to be located between 2 HAMP domains. Interestingly, both *P. aeruginosa* L3 and the L1 layer in all organisms coincide with a coupling double alpha-helix linker between two four-helical bundles. However, the composition of these layers remains unclear.”

As it is, I do not believe that the structural analysis contributes enough to the paper. However, if the authors employ 3D or 2D averaging that shows reproducible species-specific differences in array structure, then I believe the paper can be strongly considered for publication.

As suggested by the reviewer, we have added 3D averages for each organism and included these new results in a new figure and in the text.

Minor points:

- In Figure 1, it would be helpful to readers if the authors could diagram the architectural features that distinguish F6, F7, and F9 arrays. To an untrained eye, F6 and F7 look quite similar.

Thank you for this suggestion. To highlight the architectural differences, we added a panel E with the diagrams. We also improved the arrows as suggested by the reviewer #2:

We added the following to the figure legend:

“E) Diagrams of macromolecular features characteristic of the F6, F7, and F9 arrays. The F6 arrays span the inner membrane (IM) and have visible periplasmic domains (PD), and a CheA/CheW layer (AW). The F7 arrays are also membrane-bound but lack a PD. Instead, they have extra layers (EL) between the IM and the AW layer. The F9 cytoplasmic array is not bound to the membrane. Instead, the receptors are sandwiched between two AW layers.”

The labeling and figure legend of Figure 2 do not indicate that the structures being compared are all F7 arrays. This should be clearly noted.

We added the F7 code to the title of the figure in the figure legend.

- The legend of Figure 2 should explain the abbreviations IM, L3, L2, L1, SL, AW, H, P, S. Similarly, the legend of Figure 5 should explain the abbreviations PD, IM, AW, D, W, A B, R, Y7, Y6, Z6.

Thank you for these suggestions. The legend of Figure 2 now reads:

“**Fig. 2:** Structural analysis of F7 chemotaxis arrays in *P. aeruginosa*, *V. cholerae*, *S. oneidensis*, and *M. alcaliphilum*. A) Side-view of F7 chemotaxis arrays (scale bar: 50 nm) relative to the inner membrane (IM) and outer membrane (OM). B) Top-view of a sub-tomogram average at the

CheA/CheW layer reveals that F7 arrays have the typical hexagonal packing of receptors with ~12 nm spacing (scale bar: 20nm). C) Side view of the sub-tomogram averages. Several density layers (L1, L2, and L3) are present between the inner membrane and the CheA/CheW layer (AW) (scale bar: 20nm). D) A comparison of the side-view layered structure from the sub-tomogram averages with the homology models of chemoreceptors (to scale). The presence of PAS domains in the receptors structure consistently correlates with some of the layers. The box diagram shows the PFAM protein domains of each receptor: PAS (P), HAMP (H), and MCPSignal (S).”

We have also adjusted the legend for the original figure 5 (now Figure 6):

Fig. 6: Evolution of the F7 chemosensory array in non-enteric γ -Proteobacteria to acquire F6-like ultrastructure and function. A) Tomographic slices showing F6 and F7 stage 1 chemosensory arrays in the same *P. aeruginosa* cell (left) and an F7 stage 5 chemosensory array in *E. coli* (right). Over the course of evolution, the F6 system is lost and the F7 system evolves similar ultrastructure and function to the F6 system. Features to identify chemoreceptors are highlighted: Periplasmic domain (PD), Inner membrane (IM) and CheA/CheW layer (AW). B) Molecular models of F7 chemosensory arrays in *P. aeruginosa* (left) and *E. coli* (right) built based on ⁷⁷. Proteins displayed in this representation are: CheA (A), CheB(B), CheR(R), CheD(D), F7-CheY (Y₇), F6-like-CheY (Y₆) and F6-like CheZ (Z₆). Models are colored according to their hypothetical original class: F7 (red) and F6 (yellow). C) Working model of the evolution of the F7 chemosensory system in γ -Proteobacteria and β -Proteobacteria. Scale bars are 50 nm.

- The “1D electron density profiles” in Figure 2 lack a labeled X-axis.

We removed the profiles from the Figure and replaced them with sub-tomogram average densities.

Line 499: By “Gatan imaging filers” do the authors mean “Gatan energy filters”?

Done

Line 501: e-/Å² should be e-/Å².

Done

- Also in the electron cryotomography section of the methods, what was the magnification / pixel size used for acquisition?

The tomograms used in this study were taken at different magnifications and pixel sizes. We provide a complete list of identifiers of all tomograms in Table S6. These identifiers can be used to find the tomogram itself with all additional information such as pixel size and magnification on the public Electron Tomography Database (ETDB). We provide a gateway to display the tomograms from our lab on ETDB at <https://etdb.caltech.edu>.

Reviewer #2 (Remarks to the Author):

This manuscript uses bioinformatic analysis, in conjunction with information from cryo-

electron tomography, to trace the evolution of the chemotaxis system of *Escherichia coli* and related organisms. The principle findings are that this system, the paradigm for our understanding of bacterial chemosensory systems, is a hybrid of two more ancient systems and that its transmembrane chemoreceptors are chimeras generated by gene fusion. These are novel and fascinating findings that will be of interest to most involved in the study of bacterial chemotaxis and, more widely, to many involved in the study of bacterial sensory signaling. The manuscript is lucidly written. In large part, the distinction between observation and interpretation is clearly defined. However, I have one major concern and some additional comments and suggestions.

*** Major concern.

Multiple density layers. The presence of the multiple density layers besides the AW layer is not convincingly documented for the chemosensory arrays of the four species shown in Fig. 2A. At least to this reader's eyes, for *V. cholerae* none of the L layers is clearly shown in the image provided. The SL layer is obvious only for *M. alcaliphilum*, not for the other three species. For *S. oneidensis*, identification of L1 and L3 layers is unconvincing. Given these issues, the correlation the authors suggest (Fig. 2B and the relevant text in lines 171-181, 380-395) between the layers and structural features of the four receptor species would not convince a critical reader. The authors need to find a compelling way to illustrate the presence of the layers and the correlation with receptor structural features that they suggest.

This was also a concern from the Reviewer #1. We have addressed this in our responses above.

Other comments and suggestions.

1. Genes versus protein products. Throughout much of the text, the authors do a good job correctly distinguishing the two and following the convention of gene names in lower case italics; protein names in regular font with the first letter capitalized. There are a few instances where this is not the case.
 - a. Line 251. Insert "gene" after "chemoreceptor".
 - b. Line 253. tar, tap, trg and tsr should be Tar, Tap, Trg and Tsr.
 - c. Lines 280-282. The several gene names in these lines are used to modify "receptors", thus protein names should be used or "receptors" should be changed to "receptor genes".

Done

2. Lines 45-47. This compound sentence is awkward. It would read better as two separate sentences.

We have changed the line 45-47 from:

"Presumably they arose through a long series of small steps in which new components and functions accreted onto or replaced old, with"

To (now line 42):

“Presumably, they arose through a long series of small steps in which new components and functions accreted onto, or replaced, original ones. Throughout this process, each new function provided a fitness advantage and was thus retained.”

3. Line 168. Concluding from four examples that tall F7 arrays are “widespread” seems an overstatement. A revised statement would be better.

We do believe that the tall F7 arrays are widespread in γ -Proteobacteria for several reasons:

The bioinformatics analysis clearly predicts the presence of homologs of the proteins forming tall F7 arrays in almost all γ -Proteobacteria, except the enterics. Cryo-tomography is still a relatively new method and data collection is time consuming and costly. Therefore, the number of imaged γ -Proteobacteria is still not extensive. Even with the limited data availability, all species with predicted tall F7 arrays in our database (*V. cholerae*, *P. aeruginosa*, *S. oneidensis* and *M. alcaliphilum*) were confirmed to have them.

However, we do agree with the reviewer that the statement concluding this claim could be improved. Therefore, we have adjusted the text from the original version:

“Given these results in *V. cholerae*, *P. aeruginosa*, *S. oneidensis* and *M. alcaliphilum*, we conclude that the tall F7 arrays are widespread across γ -Proteobacteria.”

to

“We conclude that the tall F7 arrays are widespread across γ -Proteobacteria based on two results: (1) the bioinformatics analyses indicate the presence of homologs of the proteins forming tall F7 arrays in nearly all γ -Proteobacteria (except the enterics) and (2) the presence of tall arrays in tomograms of all four species with predicted F7 systems in our database.”

4. Line 372. I suggest using a word other than “patches” since it has been used to refer to chemoreceptor/chemosensory arrays and thus could cause confusion.

Thank you. We changed to a more specific and accurate term: alpha-helices.

5. Lines 433-457. In this text PCR primer sequences are written with lower case letters. Shouldn't upper case letters be used?

Done

6. Fig. 1. The “empty” arrows are difficult to distinguish from the black arrows when this figure is printed on a page as a hard copy. Thus it would be best to change the empty arrows to something easier to distinguish on a printed copy from the black arrows.

Thank you. We made the arrows larger to increase visibility in the new figure 1 (shown above).

Reviewer #3 (Remarks to the Author):

Previous work by the Zhulin group has shown that chemosensory pathways can be classified into flagellar systems F1-F17, Type IV pili associated functions and alternative functions. Of these systems, class F6 and F7 are particularly relevant since present in many species. Whereas F6 systems control flagellar rotation, the function of F7 systems are less well understood. Whereas the chemotaxis system of *E. coli* belongs to F7, the F7 system of *P. aeruginosa* is not involved in chemotaxis and of unknown function. The authors have conducted studies to understand the co-existence of F6 and F7 systems that differ in function.

Using electron cryotomography the authors observe two distinct chemoreceptor arrays in a number of species that differ in their width. Using different mutants the authors establish that the thinner arrays are F6 systems, whereas F7 systems form thicker arrays. Whereas canonical F6 systems contain transmembrane receptors with periplasmic/extracytosolic sensor domains the authors show that the F7 visualised by cryo-EM use soluble receptors that in many cases lack any apparent transmembrane regions. In those arrays, next to the densities caused by bound cheA/CheW, the authors observe additional layers, L1 to L3.

In the second part of this manuscript the authors establish the evolutionary history of these systems. They show that F6 and F7 systems existed contemporaneously. However, they were able to distinguish 5 different steps in the evolution of the F7 system. A key event in this evolutionary process is the recruitment of the CheY protein from an F6 pathway permitting that the output occurs at the level of the flagellar motor. Data indicate that the F7 pathways are taking over and that F6 pathways will eventually get lost. The F7 system of *P. aeruginosa* corresponds thus to an ancient systems (stage 2 in the evolution).

This manuscript is very well organized and written. The combination of cryo-EM with bioinformatics was shown to be very powerful to address a very complex issue. This manuscript provides solid answers to a number of puzzling issues in this field such as for example why some F7 systems carry out an F6 functions whereas other F7 systems have clearly a function that is different. Many bacteria have multiple chemosensory pathways and this work will permit to get initial information of these pathways.

This referee has several comments and propositions on how to improve this manuscript.

1. A significant part of literature on the *P. aeruginosa* Aer2 chemoreceptor uses with McpB a synonym. Both terms are not very fortunate since this receptor is unlikely to be an aerotaxis receptor nor a methyl-accepting chemotaxis protein. However, for clarity it should be mentioned that there is a synonym.

We introduced the synonyms in Line 131 of the original manuscript when we first mentioned this receptor (now line 136):

“In both organisms the F7 gene cluster contains two MCPs: one presumably cytosolic class 36H receptor (Aer2/McpB/PA0176 in *P. aeruginosa* and Aer2/VCA1092 in *V. cholerae*), and one receptor of uncategorized class with a predicted 132 transmembrane region (Ctp/McpA/PA0180 in *P. aeruginosa* and VCA1088 in *V. cholerae*).”

One issue has not been mentioned in this study and its assessment may provide further information into Aer2/McpB type chemoreceptors. Of the 26 *P. aeruginosa* chemoreceptors only two possess a C-terminal extensions with a terminal pentapeptide for CheR/CheB binding, which are McpB and McpA (Fig. 1 in PMID:24714571). Interestingly, both of receptors are encoded in the F7 gene cluster. In the case of McpB, this terminal pentapeptide is of crucial importance: it binds with nanomolar affinity to CheR2 and no other CheR paralogue, removal of this pentapeptides prevents it interaction with Aer2/McpB and abolishes receptor methylation. Interestingly the terminal pentapeptide of McpA appears to be some sort of degenerated: whereas the McpB pentapeptide matches the consensus defined by the Stock laboratory X-W-X-X-F, the McpA pentapeptide has the W replaced by V. In contrast to McpB, the McpA pentapeptide did not bind to any of the four CheR.

Thank you for this insight. We have added the following statement in line 309 to address this in our manuscript:

“Of the 26 chemoreceptors present in the genome of *P. aeruginosa*, only two receptors present in the F7 system gene cluster, McpA and Aer2, have a characteristic C-terminus extension³⁰. In the case of Aer2, this extension ends with a particular pentapeptide motif that is known to tether CheR2 (which is also part of the F7 gene cluster), and enhances its enzymatic activity³⁰. Our bioinformatics analysis shows that out of the 130 Aer2-like receptors in our dataset, 96 contained a matching pentapeptide tether motif: $-x-[HFYW]-x(2)-[HFYW]$ ¹⁷ (Table S11). Aer2-like sequences lacking such a peptide tether were exclusively found in genomes that contained at least one other Aer2 homolog containing a peptide tether. Therefore we conclude that this motif represents a fundamental feature of the Aer2-like family.

In contrast, the C-terminus of McpA in *P. aeruginosa* does not possess the pentapeptide motif. Instead, a valine occupies the second position of the motif. This replacement prevents an interaction with any of the 4 CheR homologs³⁰. Analysis of the other 39 McpA-like sequences in our dataset revealed that none of them possessed the $-x-[HFYW]-x(2)-[HFYW]$ motif. However, the C-terminus of McpA-like sequences is highly conserved (Fig. S4 and Table S12). The striking conservation of this region among all the McpA sequences suggests that this motif serves an important yet unclear biological role.”

Fig. S4: Sequence logo of the C-terminal of McpA-like sequences.

Table S11: End of this file.

Table S12: End of this file.

There are several questions: Is the possession of functional C-terminal pentapeptides a general feature for the family of Aer2-like receptors? Is this a feature common to receptors that feed into step 1 and step 3 F7 pathways.

Yes. As we show above, at least one of the Aer2 homologs per organism with stage 1 or 2 homologs has the C-terminal pentapeptide motif.

We did not dissect all the chemoreceptors in each genome representing the stages 1 and 2. We show above that not even all Aer2-like receptors contains the pentapeptide. Furthermore, in *E. coli*, only two out of five receptors have a pentapeptide tether. Thus, we predict that not all receptors that feed into steps 1 and 2 will have this pentapeptide motif.

Do all McpA possess a terminal pentapeptide?

We analyzed our dataset of McpAs for the presence of the pentapeptide motif. All of our 39 protein sequences contain a pentapeptide matching the -x-[VIL]-x(2)-[FY] motif. However, even though the C-terminus appears to be conserved among the homologs, its motif is different than the one proposed by Alexander and Zhulin (2007).

Is there evidence that ancestral McpA had a functional pentapeptide that during evolution has lost its capacity to bind to CheR?

No, the high conservation of the C-terminus in all McpA homologs suggests no degradation of the terminal region. Instead, it is likely required for the biological function of this receptor.

In addition, a very interesting observation is the existence of additional density layers in these F7 systems, L1 to L3. In *E. coli* CheR binds with much higher affinity to the C-terminal pentapeptide of Tar and Tsr than to receptor variants from which the pentapeptide had been removed (binding to the methylation site). In *P. aeruginosa* removal of the pentapeptide prevented CheR binding *in vitro*. The L1 layer corresponds to the zone of the C-terminal pentapeptide. Could this layer represent bound CheR/CheB? Have the authors analysed an Aer2 mutant from which the pentapeptide had been removed? The authors argue that CheR is present at low concentration. However, the Hazelbauer group has shown that CheB of *E. coli* also binds to this pentapeptide, although with lower affinity than CheR. Phosphorylation of CheB was shown to enhance the binding affinity.

We have addressed this in our response to reviewer #1. In short, we don't have enough evidence to speculate what the layers L1 and the layer L3 in *P. aeruginosa* are. A relevant excerpt of our manuscript is in line 475:

“Furthermore, previous cryo-ET of *in vitro* preparations containing only *E. coli* CheA, CheW, and Tsr showed a similar layer in that region, suggesting one or more of these proteins alone might be responsible for the L1 layer”

2. The key event in this evolutionary process is the recruitment of the F6 CheY into the F7 pathway. This incorporation needs to be accompanied by mutations in CheY enabling it to interact with the F7 CheA. Can sequence analyses provide any clues as to the changes in the F6 CheY?

Thank you for this insightful question. We gathered the CheY-F6 and the CheY-F6-like sequences in stages 3-5 and built a sequence logo for each group (new Figure 5A). To determine the changes that enable the CheY from the F6 system (CheY-F6) to interact with the CheA of the F7 system (CheA-F7), we searched for conserved positions within each group. More specifically, we searched for conserved residues between stages 3 to 5 that differ from conserved residues of CheY-F6 (marked with dots in the figure).

We then mapped these positions to the CheY NMR structure (PDB: 2LP4) in complex with the P1 and P2 domains of CheA in *E. coli* (Figure 5B). From the ten positions that matched our criteria, only two of them were not located at an interface with CheA (M45K, T56S). Three residues are facing P2 (R92K, Q94N, and V103A) and 5 are facing P1 (K22R, R26K, D27E, D37E, G62N). (Note that the numbers refer to the coordinates of the *E. coli* CheY).

Additionally, we found a charge reversal in position K117E (D in stage 4) and included it in the figure. This residue is located in the helix that is known to interact with FlhM, which is the protein-binding partner of CheY in the flagellar motor. Overall, these data support our hypothesis that the F7 system in stage 3 recruited the CheY/CheZ pair from the F6 system, and only required a small set of residue changes that are mainly located in the interface with the CheA P1-P2 domains.

New figure:

Fig. 5: Adaptation of CheY-F6 to function with F7 systems in stages 3-5. A) Comparison of the sequence logos of the CheY proteins from the F6 system and CheY-F6-like present in genomes with 3-5 F7 systems. The amino-acids are color coded according to their chemical properties: neutral (purple), polar (green), positive charge (blue), negative charge (red) and hydrophobic (black). The dots mark conserved positions within each group that are similar between stages 3 to 5 but in the group of CheY-F6. B) The alpha carbons (yellow spheres) of these positions are mapped in the structure of CheY (red) bound to the P1 and P2 domains of CheA (blue). From the ten positions that matched our criteria, only two of them were not located at an interface with CheA (M45K, T56S). Three residues are facing P2 (R92K, Q94N, and V103A) and 5 are facing P1 (K22R, R26K, D27E, D37E, G62N). Note that the numbers refer to the coordinates of the *E. coli* CheY.

We added the following passages to the main text in line 373:

“Based on our hypothesis, we predict that CheY had to acquire mutations in order to switch its interaction partner to CheA-F7. To identify these mutations, we selected the CheY protein sequences from CheY-F6 in stages 1 or 2, where the complete ancestral F7 system is still present. We then compared these sequences to those of the CheY-F6-like proteins that are predicted to interact with CheA-F7 in stages 3, 4 and 5. We summarized the sequence variability of each group using sequence logos (Fig. 5A). We found that ten positions were highly conserved in each group: They are the same in the groups with CheY-F6-like sequences (interacting with CheA-F7), but differ from those present in CheY-F6 (interacting with CheA-F6). We mapped these positions to the *E. coli* CheY NMR structure (PDB: 2LP4) in complex with the P1 and P2 domains of CheA (Fig. 5B). Only two of these positions were not located at the interface with CheA domains (M45K, T56S). From the other 8 residues, three face P2 (R92K, Q94N, and V103A) and 5 face P1 (K22R, R26K, D27E, D37E, G62N). Furthermore, we found a charge reversal in position K117E (D in stage 4, Fig. 5B). This residue is located in the helix that is known to interact with FliM, which is the protein-binding partner of CheY in the flagellar motor³². Overall, these data support our hypothesis that the F7 system in stage 3 recruited the CheY/CheZ pair from the F6 system, and this only required a small set of residue changes that are mainly located in the interface with the CheA P1-P2 domains.”

3. In stage 4 pathways there are two CheY. Has the literature on these systems been inspected to get any evidence supporting the notion that the newly incorporated CheY plays a role in chemotaxis?

Thank you for this interesting question. Indeed, a study just published in *Comamonas testosteroni* addresses this question. We have added the following text to our discussion in line 420.

“A recent study in *Comamonas testosteroni*, an organism with a stage 4 F7 system, shows that the kinase CheA is able to phosphorylate both the ancient CheY as well as the recently acquired CheY-F6-like³⁵. Deletion of the CheY-F6-like protein completely abolished chemotaxis response, while the deletion of CheY-F7 only partially affected it. The study further shows that CheY-F7 has a much faster auto-dephosphorylation rate than CheY-F6-like. The authors

interpreted these results such that the CheY-F6-like is the primary response regulator, and CheY-F7 may act as a phosphate sink. These conclusions are based on previous work in organisms with multiple CheY genes per chemosensory cluster^{36,4}.

However, because CheY-F7 in stages 1 and 2 is the sole response regulator of the system, we hypothesize that it plays a major role in the control of a yet-unknown cellular process, at least in stages 1 and 2. The rapid auto-dephosphorylation does not necessarily imply a phosphate sink as the main biological function of this CheY. Thus, in the intermediate stages of the extant β -Proteobacteria, including the system in *C. testosteroni*, the CheY-F7 may retain both the older unknown function as well as its new role in the control of the flagellar motor.

On the other hand, the immediate loss of McpA and adjacent genes from stage 1 to 3, might indicate the loss of the original F7 function. If this were true, why would the system keep a conserved response regulator (cheY) for a lost function? One hypothesis is that the original CheY and CheD might be serving in an auxiliary feedback loop in addition to the canonical CheB/CheR adaptation mechanism in stages 3 and 4. This could compensate response time of the flagellar control as the system acquired mutations to accommodate the new components and function, providing flexibility. In the stage 5, the system evolved to perform optimum response without the need of CheY/CheD adaptation mechanism, resulting in loss of these components in that stage.”

4. Both, the F6 and stage 3 -5 F7 systems mediate chemotaxis and the authors conclude that the F7 systems appear to take over. To enhance the clarity could the gene order of a typical F6 system be added to Fig. 3?

Thank you for this comment. The gene order of the F6 system is relatively uniform and has been well described in Wuichet and Zhulin. It is significantly different from the F7 system, and we feel its addition would not add to the clarity of the figure but instead make it confusing. However, in order to help the reader find the gene order information on the F6 systems, we will add the following text to the manuscript in line 257:

“The gene arrangement of F7 systems significantly differ from that of the F6 systems⁹.”

In the Discussion maybe the authors want to compare both systems and hypothesize on their potential functional advantages/disadvantages in mediating chemotaxis?

Thank you for your suggestion. Our data is not sufficient for speculating on advantages/disadvantages in mediating chemotaxis between the two systems.

Reviewer #4 (Remarks to the Author):

This manuscript describes an unusual and interesting effort to combine cryo-electron imaging, bioinformatics and evolutionary biology to describe the trajectory of a key family of bacterial sensory systems. Such cross-disciplinary efforts have the potential to make breakthroughs in problems that have proven too difficult to address by a single approach.

While the manuscript's conclusions are generally convincing, this reader was unable to

understand how the gene phylogenies were completely parallel with organismal phylogeny. Perhaps the authors simply need to make a clearer explanation to the reader that addresses the apparent inconsistencies noted in this review.

A copy of the manuscript is attached, with comments in the tracked-changes.

Ned Ruby

Thank you very much for your detailed and insightful review of our manuscript.

Many of your comments concern the temporal directionality of the gene phylogeny. These are valid points and we have therefore changed how we rooted the tree to be more formal (without changing the results). We also aimed to describe our analysis more clear to the reader.

A stronger argument should be provided the reader that makes it clear why the directionality of the evolution of the systems is assumed to be top to bottom (as indicated in Fig. 3); i.e., are the organisms that have stage 1 loci (i.e., *V. cholerae*) ancestral to those with stage 5 ? It would be useful to add (to Fig. 3) a coincident organismal phylogeny (such as in reference #28) that shows this direction of organismal descent. Unless we are mistaken, all the trees in the figures are che-gene based trees, not 16S trees.

The reviewer is correct in that all trees in the main figures are che-gene based trees. The chemosensory evolution isn't strictly congruent with the organism evolution, probably because of the large rates of lateral gene transfer in bacteria. Here we aim to focus on the evolution of the chemosensory systems and not of the organisms themselves. In particular, to the chemosensory systems of the class F7 in γ -Proteobacteria. Therefore, we built a phylogeny produced from a concatenated alignment of three chemosensory proteins, CheA, CheB and CheR (CheABR).

To give temporal directionality for the CheABR phylogeny we chose to include sequences from a sister class (F8) and root the tree on the common ancestor between the two classes. We know that F8 is a sister class of F7 based on the work from the Zhulin Lab that describes the evolutionary relationship between all chemosensory classes. We know that neither F7 or F8 are the most ancient chemosensory class based on a previously published work from our lab that found evidence that the F1 class is likely to be the oldest. Thus, we assume that the common ancestor between the classes F7 and F8 must predate all nodes from the F7 and F8 classes.

However, upon closer examination, Wuichet and Zhulin were not clear about the monophyly of the classes F7 and F8. The arrangement we originally favored (monophyletic F7 and F8 classes) was based on the CheA analysis in the Wuichet and Zhulin paper. In that same paper, the CheABR analysis placed F8 class as a sister branch of a divergent group of F7 systems called F7-divergent (F7'). In neither phylogenies, the placement of the F8 class relative to the F7 class had significant bootstrap support.

For the sake of rigor, we rebuilt our CheABR phylogeny including three sequences from the F1 class as an outgroup.

As in previous results, we were unable to confidently resolve the relationship between the classes F7 and F8 with a CheABR alignment. However, this new phylogeny shows that by rooting our original CheABR tree in any node outside the clade with sequences from F7 – stages1-5 and alpha-proteobacteria guarantees the correct temporal directionality to the part of the tree relevant to our study (F7 – stages1-5).

To avoid confusion and to keep coherent with the CheABR analysis in the Wuichet’s paper, we decided to re-root the tree in Fig 2 and S1. We included our new phylogeny of CheABR with F1 sequences as an inlet in Figure S1 marking where we chose to root the original CheABR tree (arrow). For the sake of simplicity, we decided to avoid the F7-divergent nomenclature from the Wuichet’s paper because there are no HMMs to differentiate the sequences between “regular” F7 and F7’ classes.

To make this clear to the reader, and accurate, we changed the segment in line 226 from:

“Sequences from F8 systems were included to root the tree since F8 systems share a common ancestor with F7 systems⁹”

To this (line 235):

“To give a temporal direction of evolution to this analysis, we included sequences from F8 systems to help root the tree because it is unlikely that either one of these classes is the most ancient class of chemosensory systems^{9,13}. To find an appropriate rooting point, we built an auxiliary phylogenetic inference that included sequences from the ancient class F1 as an outgroup (Figure S1-inlet). Although our analysis was unable to unambiguously determine the exact placement of the last common ancestor of the classes F7 and F8, it showed a polytomy more ancient than all F7 in both γ - and β -proteobacteria. We chose to root the CheABR phylogeny separating the F7 present in most α , γ and β -proteobacteria from the rest of the

sequences from the F7 and F8 classes, coherent with the CheABR analysis of Wuichet and Zhulin.”

In order to make the rest of the text coherent with this statement and the re-root of the tree, we also changed line 228:

“The first clade comprised all the e-proteobacterial systems”

To (line 246)

“The first clade comprised all of the F7 from e-proteobacterial, most of F7 from d-proteobacterial and all of the F8 systems”.

Line 66: Since the flagellar class seems to be an important designator, noted here and line 163, consider adding more about what distinguishes the classes; i.e., why should the reader need to know what class is being discussed.

Thank you for pointing this out. We took this opportunity to explain the evolutionary relationships relevant to this paper in more detail, and highlight the evidence that makes the class F1 the most likely to be the ancestral class of all chemosensory systems.

Original line 66:

“Chemosensory systems have been classified on the basis of evolutionary history into 17 so-called flagellar classes (F1-17), one type IV pili class (TFP), and one class of alternative cellular functions (ACF)⁹.”

We added the following for clarification:

“Because this classification system is based on phylogenetic analysis, the evolutionary relationship between the classes is generally known. Later, by analyzing chemosensory systems in archaeal genomes, we showed evidence that class F1 is the most ancient of the chemosensory classes. Understanding this classification system and its evolution allows for a temporal directionality of the evolution and diversification of this system.”

Line 83: Is this statement referring to a structural or genetic architecture?

Structural. We add the word to the sentence to avoid ambiguity. Thank you.

Line 91: hyphenate 'limited-growth'?

Done

Line 109: define heptad?

We altered the text to include the definition and a more precise description of the chemoreceptor heptad classes:

“MCPs are classified by length according to the number of heptads (sets of 7 consecutive amino-acids) that the receptors contain in their signaling domain¹⁷. The length of the shorter array (24 nm) corresponds to receptors belonging to the class with 40 heptads (40H) that are often associated with F6 systems^{18,9}, so we assign this array as the F6 system.”

Line 129: Consider replacing with 'consistent with our hypothesis'.. 'confirming' seems stronger than warranted.

Done

Line 231: The directionality of the evolution of this system is not clear. In Fig. S1, aren't there a number of beta-Proteos embedded between two gamma-Proteo clusters? Why doesn't such an arrangement indicate that the F7 stages did not descend in parallel with organismal evolution?

Yes, not only a number, but all chemosensory systems from β -Proteobacteria appear to be within γ -Proteobacteria. Our intention was to call the reader's attention to the fact that in our phylogenetic analysis, the chemosensory systems agree with the general topology of proteobacteria organisms at the class level. One exception is the inclusion of the systems present in β -Proteobacteria among the systems found in γ -Proteobacteria– as noted by the reviewer.

However, since we cannot be confident about the placement of the last common ancestor of classes F7 and F8, we decided to remove line 231:

“Thus, remarkably, the CheABR tree was mostly congruent to the general organization of Proteobacteria classes^{27,28}”

Line 232: Its stability is inferred, not established. Perhaps better to say its pattern is generally consistent with a uniform presence.

Thank you. We changed the line 232:

“This means that the F7 system has been stably associated with its cellular lineage for almost 2.8 billion years²⁸, with either very few horizontal gene transfers or the results of such shuffling events going extinct.”

To include a more accurate statement (line 249):

“The F7 chemosensory systems of the organisms from the same Proteobacteria class tend to cluster together.”

Line 233: It would be useful to point out here (or in the Fig. S2 legend) that, nevertheless, there are still several incongruences between the F7 tree (Fig. S2) and bacterial phylogeny (e.g.,

representative Shewenellaceae present in both stage 1 and 2, and Pseudomonadaceae in both stage 2 and 5 (Fig. S2), suggesting either HGT activity, or errors in the established phylogeny.

We agree that we should point out that within classes, we see evidence of HGT activity. We addressed this issue in the next comment.

Line 235: The profiles were built on what genes/genomes? Was it the same list of 31 orthologs used in reference #73? If so, please indicate, since, previously in the Results, only the CheABR sequences have been used.

Thank you for pointing this out. The genes were the same as in #31, and Table S10 contains the list of genomes used. We also think that this a good place to address the previous comment.

We changed the segment from:

“To track the distribution of the F7 system through these clades, we built a phylogenetic profile using a new random set of 162 γ -Proteobacteria, in addition to the 4 species imaged in this work and 10 β -Proteobacteria to serve as an outgroup (Fig. S2).”

To

“To track the distribution of the F7 system through these clades, we built a phylogenetic profile using a new random set of 161 γ -Proteobacteria, the 4 species imaged in this work, the model organism *E. coli*, and 10 β -Proteobacteria to serve as an outgroup (Fig. S2). The complete list of genomes is shown in Table S10, and the phylogeny was built using 31 orthologs according to the protocol Ciccarelli by et al.²⁷”

Line 241: Placing the referred-to phylogenetic tree in Fig. 2 would clarify the evidence for descent.

See our responses above. These should be sufficient to clarify how we can infer descent from a gene tree in this study.

Line 247: Perhaps best to use '5' rather than 'N', since you're talking about the orientation of the gene cluster, yes?

Done.

Line 280: Is there a gene replacement of tar from Stage 2 F6 to Stage 3 F7?

The chemoreceptors in the F6 systems are from a different heptad class than the chemoreceptors from the F7 systems. Thus, we think that *tar*-like genes from the class 36H in stages 3-5 are, in fact, a hybrid: While they contain a 36H signaling domain similar to *aer2*-like receptors, their overall topology and input domain resembles some of the 40H receptors associated with F6 systems. This remains a hypothesis that we mention in the discussion. An in-depth analysis is beyond the scope of the current manuscript.

We acknowledge that our original phrasing in the text was misleading and could lead the reader to think that F6 clusters contain a *tar*-like gene, which is not the case. Instead, 40H chemoreceptor genes with input domain and general topology similar to *tar*-like receptors are known to work with F6 systems.

To clarify, we changed the statement in line 280 from:

“In stages 1 and 2, the F7 cluster included both *mcpA*-like and *aer2*-like receptors genes, and the F6 system included a *tar*-like receptor gene.”

To (line 298):

“In stages 1 and 2, the F7 cluster contained at least one *aer2*-like receptor gene from the heptad class 36H as well as an *mcpA*-like gene. The F6 gene cluster does not contain a chemoreceptor. However, 40H receptors with a topology similar to *tar*-like genes that are known to work with F6 systems in several non-enteric γ -proteobacteria are present in these genomes.”

Line 281: Remove italics from '-like'

Done

Line 283: This statement seems like a big assumption based on a presence/absence correlation.

We toned down the segment and made it clear that this is our hypothesis based on our result.

We altered the segment from:

“Because nearly all the genomes with stage 1 and 2 F7 systems possessed *mcpA*-like and *aer2*-like chemoreceptor genes in the gene cluster, both are apparently needed for the (unknown) function of F7 in these organisms.”

To:

“Because nearly all the genomes with stage 1 and 2 F7 systems possess *mcpA*-like and *aer2*-like chemoreceptor genes in the gene cluster, we hypothesize that both receptors are needed for the yet-unknown function of F7 in these organisms.”

Line 305: transformation or gene replacement?

As we stated above, we are unsure if it is the case of gene replacement or domain swap. Therefore, we changed the text as follows:

From:

“The change in the biological function of the F7 system apparently coincided with the transformation of the *aer2*-like gene into a *tar*-like gene.”

To:

“The change in the biological function of the F7 system apparently coincided with the change of the *aer2*-like gene to a *tar*-like gene.”

Line 311 - Is this tar?

No. We understand that the issue in Line 280 addressed above might have led the reviewer to conclude that F6 systems had *tar*-like receptors. We meant that F6 systems have chemoreceptors with the same topology as *tar*-like receptors, but they are from a different heptad class.

Line 314: remove 'that'

Done

Line 321: Two genes, three alleles?

Our original statement was confusing.

We changed it from:

“These two genes were retained in stages 3 and 4, even as the F6 cluster was lost.”

To:

“The *cheY/cheZ* pair from the F6 systems was retained in stages 3 and 4, even as the remainder of the F6 cluster was lost.”

Line 326: If both *cheY*s are present in Stage 1, how is it possible to judge which is older?

We meant that it is older in the F7 gene cluster. We changed the segment to clarify this information from:

“This showed that the “extra” *cheY* genes outside the F7 gene cluster in stage 3 genomes were more closely related to F6 *cheY*s than to the older F7 *cheY*s.”

To:

“This shows that the “extra” *cheY* genes outside the F7 gene cluster in stage 3 genomes were more closely related to F6 *cheY* genes than to the *cheY* genes present in F7 stage 1 and 2.”

Line 350: 'eventual' rather than 'gradual'.. there's no evidence presented concerning the rate.

Done

Line 389: This sentence seems to change direction midway. Perhaps reformatting to read: 'It is unlikely that this density is produced by another protein. For example, while CheR is known to bind the chemoreceptor in that area, the abundance of this protein....'

We altered this sentence to address the reviewer's #1 concern.

Line 667: Please indicate which figure contains the 'organismal tree'.

Done.

Line 755: The references are not uniform in format, and have errors (e.g., the date in #74).

Done.

Line 984: Please identify the abbreviations used in the figure (e.g., SL, AW).

Done.

Line 997: 'gene'

Done

Figure 4:

- 1) The genus names of these *C. testosteroni* and *B. bronchiseptica* are not identified anywhere.
- 2) *C. testosteroni* and *B. bronchiseptica* don't seem to be described anywhere in the text.

We picked those two representative genomes to show the placement of cheYs from stages 3 and 4.

We add the following sentence to the legend:

The representative genomes for each stage are: Stage 1: *Vibrio cholerae*, Stage 2: *Pseudomonas aeruginosa*, Stage 3: *Bordetella bronchiseptica*, Stage 4: *Comamonas testosteroni* and Stage5: *Escherichia coli*.

Figure 5:

use lower-case letters so as not to confuse with panel designations

Thank you for pointing this out. We decided to make the panel letters bold and larger.

Table S11: Aer2-like pentapeptide tethers.

Organism identifier	locus	accession	pentapeptide
Al_mac_7736	I876_01970	YP_008194818.1	DWEAF
Al_mac_7736	I876_02010	YP_008194826.1	EWETF

Organism identifier	locus	accession	pentapeptide
Al_mac_7736	I876_02015	YP_008194827.1	EWESF
Al_sp._1413	ambt_16735	YP_004468654.1	EWEAF
Al_vin_90	Alvin_0183	YP_003442182.1	QWEEF
Al_vin_90	Alvin_1872	YP_003443828.1	
Al_vin_90	Alvin_2222	YP_003444173.1	
Al_vin_90	Alvin_2230	YP_003444181.1	
Gl_nit_1515	GNIT_1657	YP_004871766.1	EWKEF
Gl_sp._1395	Glaag_2576	YP_004434785.1	EWESF
Ha_che_746	HCH_00457	YP_431792.1	DWEVF
Ha_che_746	HCH_00458	YP_431793.1	DWEVF
Ha_hal_741	Hhal_2163	YP_001003729.1	EWEEF
Li_ang_7812	N175_16910	YP_008489689.1	EWEEF
Ma_med_1360	Marme_1102	YP_004312213.1	DWEEF
Ma_sp._859	Mmwyl1_3301	YP_001342141.1	GWEEF
Me_alc_1536	MEALZ_2872	YP_004918123.1	EWEEF
Me_met_1418	Metme_2154	YP_004513058.1	EWQDF
Ps_aer_479	PA0176	NP_248866.1	GWEEF
Ps_aer_7891	PA1S_gp3690	REF_DMTMMU:PA1S_gp3690	GWEEF
Ps_den_2356	H681_00805	YP_007655583.1	DWEEF
Ps_res_7713	PCA10_13880	YP_008101725.1	EWEEF
Ps_sp._1241	PSM_A2954	YP_004070018.1	EWEEF
Ps_suw_1301	Psesu_0059	YP_004145153.1	DWQEF
Ps_suw_1301	Psesu_1463	YP_004146541.1	
Ps_suw_1301	Psesu_1465	YP_004146543.1	EWAKF
Ps_suw_1301	Psesu_1466	YP_004146544.1	DWAEF
Sa_deg_468	Sde_3105	YP_528574.1	DWEDF
Sh_ama_634	Sama_3497	YP_929369.1	EWHEF
Sh_bal_241	Sbal175_2162	YP_006020732.1	EWEEF
Sh_loi_680	Shew_0111	YP_001092242.1	EWNEF

Organism identifier	locus	accession	pentapeptide
Sh_one_481	SO_2123	NP_717726.1	EWEEF
Sh_sed_917	Ssed_0184	YP_001471925.1	EWNEF
Sh_sp._679	Shewana3_2216	YP_869851.1	EWEDF
Sh_vio_130	SVI_0176	YP_003554925.1	EWNEF
Sh_woo_862	Swoo_0164	YP_001758560.1	EWNEF
Si_aga_2165	M5M_00415	YP_006915050.1	EWEEF
St_mal_1491	BurJV3_1158	YP_004791716.1	GWEEF
St_mal_1491	BurJV3_1903	YP_004792454.1	DWQEF
St_mal_1491	BurJV3_1904	YP_004792455.1	DWQEF
St_mal_1491	BurJV3_1908	YP_004792459.1	DWQEF
St_mal_1491	BurJV3_2459	YP_004793006.1	
St_mal_1491	BurJV3_3037	YP_004793581.1	
St_mal_1491	BurJV3_3580	YP_004794119.1	
St_mal_1491	BurJV3_3943	YP_004794481.1	
Te_tur_1125	TERTU_1341	YP_003072897.1	EWEDF
Te_tur_1125	TERTU_2935	YP_003074319.1	
Th_cru_598	Tcr_0553	YP_390823.1	DWSDF
Th_cru_598	Tcr_2004	YP_392268.1	
Th_ole_2361	TOL_2508	YP_007683144.1	EWEEF
Th_vio_1521	Thivi_0439	YP_006412631.1	QWEEF
Th_vio_1521	Thivi_1211	YP_006413359.1	DWEEF
Th_vio_1521	Thivi_1222	YP_006413370.1	EWSEF
Vi_ang_1433	VAA_01905	YP_004577835.1	EWEEF
Vi_cho_1795	VC395_0082	YP_002818346.1	EWESF
Vi_cho_1795	VC395_A1113	YP_002822179.1	EWEEF
Vi_cho_319	VC0098	NP_229757.1	EWESF
Vi_cho_319	VCA1092	NP_233472.1	EWEEF
Vi_fur_1564	vfu_B00980	YP_005049501.1	EWEEF
Vi_nig_7850	VIBNI_B0011	YP_008640924.1	EWEEF

Organism identifier	locus	accession	pentapeptide
Vi_nig_7850	VIBNI_B0830	YP_008641677.1	
Vi_vul_1326	VV2_1165	NP_763073.1	EWEEF
Xa_alb_65	XALc_0649	YP_003375155.1	DWEEF
Xa_alb_65	XALc_1357	YP_003375852.1	
Xa_alb_65	XALc_1361	YP_003375856.1	QWRDF
Xa_alb_65	XALc_1362	YP_003375857.1	HWHEF
Xa_alb_65	XALc_1364	YP_003375859.1	QWQEF
Xa_alb_65	XALc_1365	YP_003375860.1	SWQEF
Xa_alb_65	XALc_1926	YP_003376405.1	
Xa_alb_65	XALc_2151	YP_003376626.1	NWQEF
Xa_alb_65	XALc_2152	YP_003376627.1	DWQEF
Xa_alb_65	XALc_2153	YP_003376628.1	DWQEF
Xa_alb_65	XALc_3131	YP_003377604.1	
Xa_axo_1502	XACM_0614	YP_004850217.1	
Xa_axo_1502	XACM_1288	YP_004850870.1	DWQDF
Xa_axo_1502	XACM_1685	YP_004851263.1	
Xa_axo_1502	XACM_1913	YP_004851485.1	NWQEF
Xa_axo_1502	XACM_1918	YP_004851490.1	
Xa_axo_1502	XACM_1920	YP_004851492.1	NWQEF
Xa_axo_1502	XACM_1921	YP_004851493.1	DWQEF
Xa_axo_1502	XACM_1922	YP_004851494.1	
Xa_axo_1502	XACM_1923	YP_004851495.1	SWQEF
Xa_axo_1502	XACM_1925	YP_004851496.1	NWAEF
Xa_axo_1502	XACM_1926	YP_004851497.1	DWSEF
Xa_axo_1502	XACM_1927	YP_004851498.1	QWQDF
Xa_axo_1502	XACM_1929	YP_004851500.1	QWQDF
Xa_axo_1502	XACM_1930	YP_004851501.1	
Xa_axo_1502	XACM_1932	YP_004851503.1	NWQEF
Xa_axo_1502	XACM_1933	YP_004851504.1	SWQEF

Organism identifier	locus	accession	pentapeptide
Xa_axo_1502	XACM_3051	YP_004852602.1	
Xa_cam_666	XCV0669	YP_362400.1	
Xa_cam_666	XCV1702	YP_363433.1	
Xa_cam_666	XCV1933	YP_363664.1	NWQEF
Xa_cam_666	XCV1938	YP_363669.1	
Xa_cam_666	XCV1939	YP_363670.1	NWQEF
Xa_cam_666	XCV1940	YP_363671.1	DWQEF
Xa_cam_666	XCV1941	YP_363672.1	
Xa_cam_666	XCV1942	YP_363673.1	SWQEF
Xa_cam_666	XCV1944	YP_363675.1	SWQEF
Xa_cam_666	XCV1945	YP_363676.1	NWAEF
Xa_cam_666	XCV1947	YP_363678.1	DWSEF
Xa_cam_666	XCV1948	YP_363679.1	QWQDF
Xa_cam_666	XCV1951	YP_363682.1	QWQDF
Xa_cam_666	XCV1952	YP_363683.1	
Xa_cam_666	XCV1954	YP_363685.1	NWQEF
Xa_cam_666	XCV1955	YP_363686.1	SWQEF
Xa_cam_666	XCV3261	YP_364992.1	
Xa_cit_2353	XCAW_02407	YP_007650389.1	
Xa_cit_2353	XCAW_02490	YP_007650471.1	DWQEF
Xa_cit_2353	XCAW_02492	YP_007650473.1	
Xa_cit_2353	XCAW_02493	YP_007650474.1	QWQDF
Xa_cit_2353	XCAW_02495	YP_007650476.1	QWQDF
Xa_cit_2353	XCAW_02496	YP_007650477.1	DWSEF
Xa_cit_2353	XCAW_02497	YP_007650478.1	NWAEF
Xa_cit_2353	XCAW_02498	YP_007650479.1	SWQEF
Xa_cit_2353	XCAW_02499	YP_007650480.1	SWQEF
Xa_cit_2353	XCAW_02500	YP_007650481.1	
Xa_cit_2353	XCAW_02501	YP_007650482.1	NWQEF

Organism identifier	locus	accession	pentapeptide
Xa_cit_2353	XCAW_02502	YP_007650483.1	NWQEF
Xa_cit_2353	XCAW_02504	YP_007650485.1	
Xa_cit_2353	XCAW_02508	YP_007650489.1	NWQEF
Xa_cit_2353	XCAW_03417	YP_007651390.1	
Xa_cit_2353	XCAW_03970	YP_007651933.1	
Xa_cit_2353	XCAW_04466	YP_007652428.1	
Xa_ory_584	XOO2840	YP_201479.1	DWAEF
Xa_ory_584	XOO2842	YP_201481.6	NWAEF
Xa_ory_584	XOO2844	YP_201483.1	SWQEF
Xa_ory_584	XOO2845	YP_201484.1	
Xa_ory_584	XOO2847	YP_201486.1	NWQDF
Xa_ory_584	XOO2848	YP_201487.6	DWQDF

Table S12: McpA-like C-terminal motif

Organism identifier	locus	accession	pentapeptide
Al_mac_7736	I876_01990	YP_008194822.1	DIELF-
Al_mac_7736	I876_01950	YP_008194814.1	EVELF-
Al_sp._1413	ambt_16755	YP_004468658.1	EVELF-
Gl_sp._1395	Glaag_2582	YP_004434791.1	DLELF-
Ma_med_1360	Marme_1096	YP_004312207.1	EIDLF-
Ma_sp._859	Mmwyl1_3295	YP_001342135.1	DIDLF-
Sh_sp._679	Shewana3_2222	YP_869857.1	EIELF-
Sh_one_481	SO_2117	NP_717720.2	EIELF-
Sh_bal_241	Sbal175_2156	YP_006020726.1	EIELF-
Ps_den_2356	H681_00825	YP_007655587.1	EVELF-
Ps_res_7713	PCA10_13920	YP_008101729.1	EVELF-
Ps_aer_479	PA0180	NP_248870.1	EVELF-
Ps_aer_7891	PA1S_gp3694	REF_DMTMMU:PA1S_gp3694	EVELF-
Th_cru_598	Tcr_0759	YP_391029.1	EIDLF-

Organism identifier	locus	accession	pentapeptide
Ha_hal_741	Hhal_2159	YP_001003725.1	DVELF-
Gl_nit_1515	GNIT_1661	YP_004871770.1	DIELF-
Me_alc_1536	MEALZ_2878	YP_004918129.1	DIELF-
Me_met_1418	Metme_2161	YP_004513065.1	DVELF-
Fr_aur_1557	Fraau_2042	YP_005378108.1	DIDLF-
Rh_sp._1523	R2APBS1_2961	YP_007591271.1	EIELF-
Al_vin_90	Alvin_2234	YP_003444185.1	DIELF-
Th_vio_1521	Thivi_1219	YP_006413367.1	DIELF-
Ha_che_746	HCH_00449	YP_431784.1	DIELF-
Sa_deg_468	Sde_3111	YP_528580.1	EIVLY-
Te_tur_1125	TERTU_1204	YP_003072781.1	DIELYE
Ps_suw_1301	Psesu_1530	YP_004146608.1	TVELF-
Xa_alb_65	XALc_1440	YP_003375935.1	TVELF-
St_mal_1491	BurJV3_1974	YP_004792525.1	TVELF-
Xa_ory_584	XOO2558	YP_201197.1	TVELF-
Xa_cit_2353	XCAW_01830	YP_007649819.1	TVELF-
Xa_axo_1502	XACM_2022	YP_004851592.1	TVELF-
Xa_cam_666	XCV2044	YP_363775.1	TVELF-
Vi_vul_1326	VV2_1160	NP_763069.1	EVELF-
Vi_nig_7850	VIBNI_B0016	YP_008640928.1	EVELF-
Vi_fur_1564	vfu_B00976	YP_005049497.1	EVELF-
Vi_cho_1795	VC395_A1109	YP_002822175.1	EVELF-
Vi_cho_319	VCA1088	NP_233469.1	EVELF-
Vi_ang_1433	VAA_01909	YP_004577831.1	EVELF-
Li_ang_7812	N175_16890	YP_008489685.1	EVELF-

REVIEWERS' COMMENTS:

Reviewer #1 (Remarks to the Author):

The authors address my major concerns with their revised manuscript. I appreciate the effort to perform subtomogram averaging of the F7 arrays from all four species, which I think has strengthened the structural claims made in this paper.

Some small issues remain with Figure 2: For some reason, the layer densities look much clearer to me in Fig. 2C than in 2D (where the comparison to homology models is made). Hopefully, the authors can improve that. Also, in Fig. 2C for *S. oneidensis*, the labels for L2 and L3 do not match up with the densities. This should be fixed.

As noted by the authors, the subtomogram analysis reveals additional L1 and L3 densities that do not match the homology models. This is ok, and importantly, the authors include commentary on these extra densities in their revised manuscript. Every effort should be made to clarify this for general audience readers to prevent confusion.

From a structural standpoint, this manuscript is much improved. I find myself wishing for even more clear structural analysis, but I appreciate that the current datasets are limited. I believe this manuscript is suitable for publication.

Reviewer #2 (Remarks to the Author):

In the revised version of this manuscript and their accompanying comments, the authors have adequately addressed the concerns and suggestions I expressed about the initial version.

However, I strongly encourage the authors to perform a detailed and thorough checking of every aspect of this complex and extensive piece manuscript for accuracy and validity, correcting errors and mistakes that are found. Two examples of the need for correction that I happened to notice (there may well be many more):

1. Fig. 2.C in the image of the *S. oneidensis* array, the labels are not aligned with the respective layers.
2. In the paragraph beginning on line 420 that summarizes observations about CheY's from *Comomonas testosteroni*, the reference cited on line 422, ref. 36, has nothing to do with CheY's in that organism. The authors probably meant to cite another 2019 paper about *Comomonas testosteroni*, Huang et al. 2019 mBio. Insuring accurate citations seems particularly important for a multi-faceted study like this one.

Reviewer #3 (Remarks to the Author):

This is a nice piece of work. The additions have significantly improved the manuscript.

Reviewer #4 (Remarks to the Author):

Lines 67-71: Actually, the system is based on 'phylogenomics', not phylogenetics. There doesn't seem to be any direct linkage made between the cheA-cheB-cheR phylogeny and genomic phylogeny in Wuichet 2010.

Line 232: Generally, evolutionary/descent inferences are based on gene sequences rather than

protein sequences due to the degeneracy of the genetic code, and the loss of information when translating gene sequences to protein ones. It would be useful to know how well phylogenies made from these two resemble each other. It seems likely that better evidence for the evolutionary relationship between CSS stages would be determined from a concatenated tree of cheA-cheB-cheR genes (and/or three separate trees for each gene, which should be largely congruent, as well).

Supplemental, Line 50-51: There seems to be something wrong with this sentence, or else its meaning is opaque. Probably just needs the grammar tweaked. "Phylogenetic profile of the F7 and F6 systems in γ -Proteobacteria shows that only organisms with stage 1 (red) and from stage 2 (green) has F6 systems but not from stage 5 (blue)."

Supplemental, Line 68: 'cholerae'

Reviewer #1 (Remarks to the Author):

The authors address my major concerns with their revised manuscript. I appreciate the effort to perform subtomogram averaging of the F7 arrays from all four species, which I think has strengthened the structural claims made in this paper.

Some small issues remain with Figure 2: For some reason, the layer densities look much clearer to me in Fig. 2C than in 2D (where the comparison to homology models is made). Hopefully, the authors can improve that. Also, in Fig. 2C for *S. oneidensis*, the labels for L2 and L3 do not match up with the densities. This should be fixed.

Thank you for your suggestion. We also agree with the reviewer that Fig. 2C looks much clearer. This is precisely the reason why we built panel C. This is a well-known problem when zooming in electron micrographs. We have tried our best to increase contrast.

Thank you for calling our attention to the labels in Fig 2C, we fixed the *S. oneidensis* and the *V. cholerae*.

As noted by the authors, the subtomogram analysis reveals additional L1 and L3 densities that do not match the homology models. This is ok, and importantly, the authors include commentary on these extra densities in their revised manuscript. Every effort should be made to clarify this for general audience readers to prevent confusion.

We are glad that the reviewer is satisfied with our approach.

From a structural standpoint, this manuscript is much improved. I find myself wishing for even more clear structural analysis, but I appreciate that the current datasets are limited. I believe this manuscript is suitable for publication.

Thank you.

Reviewer #2 (Remarks to the Author):

In the revised version of this manuscript and their accompanying comments, the authors have adequately addressed the concerns and suggestions I expressed about the initial version.

However, I strongly encourage the authors to perform a detailed and thorough checking of every aspect of this complex and extensive piece manuscript for accuracy and validity, correcting errors and mistakes that are found. Two examples of the need for correction that I happened to notice (there may well be many more):

1. Fig. 2.C in the image of the *S. oneidensis* array, the labels are not aligned with the respective layers.

We addressed this as per request of the Reviewer #1.

2. In the paragraph beginning on line 420 that summarizes observations about CheY's from *Comomonas testosteroni*, the reference cited on line 422, ref. 36, has nothing to do with CheY's in that organism. The authors probably meant to cite another 2019 paper about *Comomonas testosteroni*, Huang et al. 2019 mBio. Insuring accurate citations seems particularly important for a multi-faceted study like this one.

The reviewer is correct and we meticulously re-checked our references.

Reviewer #3 (Remarks to the Author):

This is a nice piece of work. The additions have significantly improved the manuscript.
Thank you

Reviewer #4 (Remarks to the Author):

Lines 67-71: Actually, the system is based on 'phylogenomics', not phylogenetics. There doesn't seem to be any direct linkage made between the cheA-cheB-cheR phylogeny and genomic phylogeny in Wuichet 2010.

We correct the term. Thank you.

Line 232: Generally, evolutionary/descent inferences are based on gene sequences rather than protein sequences due to the degeneracy of the genetic code, and the loss of information when translating gene sequences to protein ones. It would be useful to know how well phylogenies made from these two resemble each other. It seems likely that better evidence for the evolutionary relationship between CSS stages would be determined from a concatenated tree of cheA-cheB-cheR genes (and/or three separate trees for each gene, which should be largely congruent, as well).

The reviewer is correct that in evolutionary analysis of closely related systems with a relatively recent common ancestor, for example, genes common to primates, birds, etc., a phylogeny using DNA sequence rather than protein sequences are more informative. However, for distantly related systems, such as the F7 chemosensory systems are in all Gammaproteobacteria, we think it is more appropriate to use protein sequences. In these cases, loss of information does not come only from translating gene sequences to proteins. Instead, the large number of mutations in a single nucleotide position over the long evolutionary history covered in our analysis misleads maximum likelihood inferences.

We are not the first ones to use this technique. Previously, others have used concatenated alignments of proteins to infer the evolutionary history of organisms for example in Ciccarelli, F. D., et. al., *Science* (2005) and Parks, D. H., et al., *Nat. Biotechnology* (2018)). In 2010, Wuichet and Zhulin used concatenated alignment of

CheA-CheB-CheR proteins to determine the evolutionary history of chemosensory pathways. In that paper, they also examine individual CheA, CheB, and CheR trees and concluded that they are mostly congruent. We think that based on those standards, we did not need to show the congruency among the alignments for each protein family, as our analysis is a more in-depth version of the work published by Wuichet and Zhulin (2010).

Supplemental, Line 50-51: There seems to be something wrong with this sentence, or else its meaning is opaque. Probably just needs the grammar tweaked. “Phylogenetic profile of the F7 and F6 systems in g-Proteobacteria shows that only organisms with stage 1 (red) and from stage 2 (green) has F6 systems but not from stage 5 (blue).”

Thank you, we changed to:

“Phylogenetic profile of the F7 and F6 systems in g-Proteobacteria shows that only organisms with F7 stage 1 (red) or stage 2 (green) systems also has F6 systems. Organisms with F7 stage 5 (blue) does not have F6 systems.”

Supplemental, Line 68: ‘cholerae’

Done